# Structurally distributed surface sites tune allosteric regulation

**James W McCormick**[1,2], **Marielle AX Russo**[1,2], **Samuel Thompson**[3],
**Aubrie Blevins**[1], **Kimberly A Reynolds**[1,2]*

[1]The Green Center for Systems Biology, University of Texas Southwestern Medical Center, Dallas, United States; [2]Department of Biophysics, University of Texas Southwestern Medical Center, Dallas, United States; [3]Department of Bioengineering, Stanford University, Stanford, United States

**Abstract** Our ability to rationally optimize allosteric regulation is limited by incomplete knowledge of the mutations that tune allostery. Are these mutations few or abundant, structurally localized or distributed? To examine this, we conducted saturation mutagenesis of a synthetic allosteric switch in which Dihydrofolate reductase (DHFR) is regulated by a blue-light sensitive LOV2 domain. Using a high-throughput assay wherein DHFR catalytic activity is coupled to *E. coli* growth, we assessed the impact of 1548 viable DHFR single mutations on allostery. Despite most mutations being deleterious to activity, fewer than 5% of mutations had a statistically significant influence on allostery. Most allostery disrupting mutations were proximal to the LOV2 insertion site. In contrast, allostery enhancing mutations were structurally distributed and enriched on the protein surface. Combining several allostery enhancing mutations yielded near-additive improvements to dynamic range. Our results indicate a path toward optimizing allosteric function through variation at surface sites.

## Introduction

In allosteric regulation, protein activity is modulated by an input effector signal spatially removed from the active site. Allostery is a desirable engineering target because it can yield sensitive, reversible, and rapid control of protein activity in response to diverse inputs (*Dagliyan et al., 2019*; *Pincus et al., 2017*; *Raman et al., 2014*). One common approach for achieving allosteric regulation in both engineered and evolved systems is through domain insertion: the transposition, recombination, or otherwise fusion of an 'input' domain into an 'output' domain of interest (*Aroul-Selvam et al., 2004*; *Dagliyan et al., 2016*; *Ostermeier and Benkovic, 2000*; *Nadler et al., 2016*). In natural proteins, domain insertions and rearrangements play a key role in generating regulatory diversity, with kinases serving as a prototypical example (*Fan et al., 2018*; *Huse and Kuriyan, 2002*; *Peisajovich et al., 2010*; *Shah et al., 2018*). In engineered proteins, domain insertions have been used to generate fluorescent metabolite biosensors (*Nadler et al., 2016*), sugar-regulated TEM-1 β-lactamase variants (*Guntas et al., 2005*), and a myriad of light-controlled proteins including kinases, ion channels, guanosine triphosphatases, guanine exchange factors, and Cas9 variants (*Dagliyan et al., 2016*; *Wang et al., 2016*; *Karginov et al., 2011*; *Toettcher et al., 2013*; *Shaaya et al., 2020*; *Coyote-Maestas et al., 2019*; *Richter et al., 2016*). In all cases, domain insertion provides a powerful means to confer new regulation in a modular fashion.

However, naively created domain insertion chimeras sometimes exhibit relatively modest allosteric dynamic range, with small observed differences in activity between the constitutive and activated states (*Lee et al., 2008*). These fusions then require further optimization by either evolution or empirical mutagenesis, but general principles to guide this process are largely absent. Which mutations tune or improve an allosteric system? Because we lack comprehensive studies of allosteric

*For correspondence:
kimberly.reynolds@
utsouthwestern.edu

Competing interests: The authors declare that no competing interests exist.

**eLife digest** Many proteins exhibit a property called 'allostery'. In allostery, an input signal at a specific site of a protein – such as a molecule binding, or the protein absorbing a photon of light – leads to a change in output at another site far away. For example, the protein might catalyze a chemical reaction faster or bind to another molecule more tightly in the presence of the input signal. This protein 'remote control' allows cells to sense and respond to changes in their environment. An ability to rapidly engineer new allosteric mechanisms into proteins is much sought after because this would provide an approach for building biosensors and other useful tools. One common approach to engineering new allosteric regulation is to combine a 'sensor' or input region from one protein with an 'output' region or domain from another.

When researchers engineer allostery using this approach of combining input and output domains from different proteins, the difference in the output when the input is 'on' versus 'off' is often small, a situation called 'modest allostery'. McCormick et al. wanted to know how to optimize this domain combination approach to increase the difference in output between the 'on' and 'off' states.

More specifically, McCormick et al. wanted to find out whether swapping out or mutating specific amino acids (each of the individual building blocks that make up a protein) enhances or disrupts allostery. They also wanted to know if there are many possible mutations that change the effectiveness of allostery, or if this property is controlled by just a few amino acids. Finally, McCormick et al. questioned where in a protein most of these allostery-tuning mutations were located.

To answer these questions, McCormick et al. engineered a new allosteric protein by inserting a light-sensing domain (input) into a protein involved in metabolism (a metabolic enzyme that produces a biomolecule called a tetrahydrofolate) to yield a light-controlled enzyme. Next, they introduced mutations into both the 'input' and 'output' domains to see where they had a greater effect on allostery.

After filtering out mutations that destroyed the function of the output domain, McCormick et al. found that only about 5% of mutations to the 'output' domain altered the allosteric response of their engineered enzyme. In fact, most mutations that disrupted allostery were found near the site where the 'input' domain was inserted, while mutations that enhanced allostery were sprinkled throughout the enzyme, often on its protein surface. This was surprising in light of the commonly-held assumption that mutations on protein surfaces have little impact on the activity of the 'output' domain. Overall, the effect of individual mutations on allostery was small, but McCormick et al. found that these mutations can sometimes be combined to yield larger effects.

McCormick et al.'s results suggest a new approach for optimizing engineered allosteric proteins: by introducing mutations on the protein surface. It also opens up new questions: mechanically, how do surface sites affect allostery? In the future, it will be important to characterize how combinations of mutations can optimize allosteric regulation, and to determine what evolutionary trajectories to high performance allosteric 'switches' look like.

mutational effects in either engineered or natural systems, it remains unclear whether such mutations are common or rare, and what magnitude of allosteric effect we might typically expect for single mutations. Additionally, it is not obvious if such mutations are structurally distributed or localized (for example, to the insertion site). Answers to these questions would inform practical strategies for optimizing engineered systems and provide insight into the evolution of natural multi-domain regulation in proteins.

To address these questions, we performed a deep mutational scan of a synthetic allosteric switch: a fusion between the *E. coli* metabolic enzyme Dihydrofolate Reductase (DHFR) and the blue-light sensing LOV2 domain from *A. sativa* (**Lee et al., 2008**; **Reynolds et al., 2011**). This modestly allosteric chimera shows a 30% increase in DHFR velocity in response to light. Focusing on mutations to the DHFR residues, we found that only a small fraction (4.4%) of the mutations that retained DHFR activity had a statistically significant impact on allostery. Individual mutations exhibited generally modest effect sizes; the most allosteric single mutant characterized (H124Q) yielded a twofold increase in velocity in response to light relative to the starting construct. Structurally, allostery

disrupting mutations tended to cluster near the LOV2 insertion site and were modestly enriched at both conserved and co-evolving amino acid positions. In contrast, allostery enhancing mutations were distributed across the protein, and strongly associated with the protein surface. We observed that combining a few of these mutations yielded near-additive enhancements to allosteric dynamic range. Collectively, our data elucidates practical strategies for optimizing engineered systems, and shows that weakly conserved, structurally distributed surface sites can contribute to allosteric tuning.

## Results

### Characterization of an unoptimized allosteric fusion of DHFR-LOV2

To begin our study of allostery tuning mutations, we selected a previously characterized synthetic allosteric fusion between DHFR and LOV2 generated in earlier work (*Lee et al., 2008*; *Reynolds et al., 2011*). In this fusion, the LOV2 domain of *A. sativa* is inserted between residues 120 and 121 of the *E. coli* DHFR βF-βG loop; we refer to this construct as DL121 (*Figure 1A,B*). The choice of LOV2 insertion site was guided by Statistical Coupling Analysis (SCA), an approach for analyzing coevolution between pairs of amino acids across a homologous protein family (*Rivoire et al., 2016*; *Lockless and Ranganathan, 1999*; *Halabi et al., 2009*). A central finding of SCA is that co-evolving groups of amino acids, termed *sectors*, often form physically contiguous networks in the tertiary structure that link allosteric sites to active sites (*Halabi et al., 2009*; *Süel et al., 2003*; *Pincus et al., 2018*). To create the DL121 fusion, Lee et al. followed the guiding principle that sector connected surface sites in DHFR might serve as preferred sites (or 'hot spots') for the introduction of allosteric regulation (*Lee et al., 2008*). The resulting DL121 fusion covalently attaches the N- and C-termini of LOV2 into a sector connected surface on DHFR, and displays a twofold increase in DHFR hydride transfer rate ($k_{hyd}$) in response to blue light (*Lee et al., 2008*). Under steady-state conditions, we measured a 28% increase in the turnover number ($k_{cat}$) in response to light and a statistically insignificant change in the Michaelis constant ($K_m$) (*Figure 1C*). Thus, the DL121 fusion is modestly allosteric in vitro. As DHFR has no known natural allosteric regulation, the LOV2 insertion confers a new, evolutionarily unoptimized regulatory input.

But can this relatively small allosteric effect generate measurable physiological differences that could provide the basis for evolutionary selection? DHFR catalyzes the reduction of 7,8-dihydrofolate (DHF) to 5,6,7,8-tetrahydrofolate (THF) using NADPH as a co-factor. THF then serves as a one-carbon donor and acceptor in the synthesis of thymidine, purine nucleotides, serine, glycine, and methionine. Because of these critical metabolic functions, DHFR activity is strongly linked to growth rate, and under appropriate conditions, *E. coli* growth rate can be used as a proxy for DHFR activity (*Reynolds et al., 2011*; *Thompson et al., 2020*). Prior work found that the modest in vitro allosteric effect of DL121 conferred a selectable growth rate advantage in vivo: when an *E. coli* DHFR deletion strain (ER2566 Δ*folA*Δ*thyA*) was complemented with DL121, the resulting strain grew 17% faster in the light than in the dark (*Reynolds et al., 2011*). Thus, DL121 is a system where: (1) allosteric control is rapidly and reversibly applied, (2) the allosteric effects on activity can be readily quantified both in vitro and in vivo, and (3) there remains potential for large improvements in regulatory dynamic range through mutation.

### A high-throughput assay to resolve small changes in DHFR catalytic activity

Our goal was to measure the effect of every single amino acid mutation in DHFR on the allosteric regulation of DL121. To do this, we aimed to follow a strategy loosely akin to a double mutant cycle (*Figure 1D*). The starting DL121 construct shows so-called V-type allostery, in which the effector (light) regulates the catalytic turnover number ($k_{cat}$) (*Carlson and Fenton, 2016*). Thus, allostery can be quantified as the ratio of $k_{cat}$ between lit and dark states. More generally, allostery might be considered as a ratio of velocities ($v = k_{cat} [S]/(K_m + [S])$) between the lit and dark states, as the allosteric effector could regulate turnover, substrate affinity, or both. In either case, we defined the allosteric effect of mutation as the fold change in allosteric regulation upon mutation (*Figure 1D*, blue box). We sought to infer this quantity for every mutation in a saturation mutagenesis library of DHFR by using growth rate as a proxy for catalytic activity.

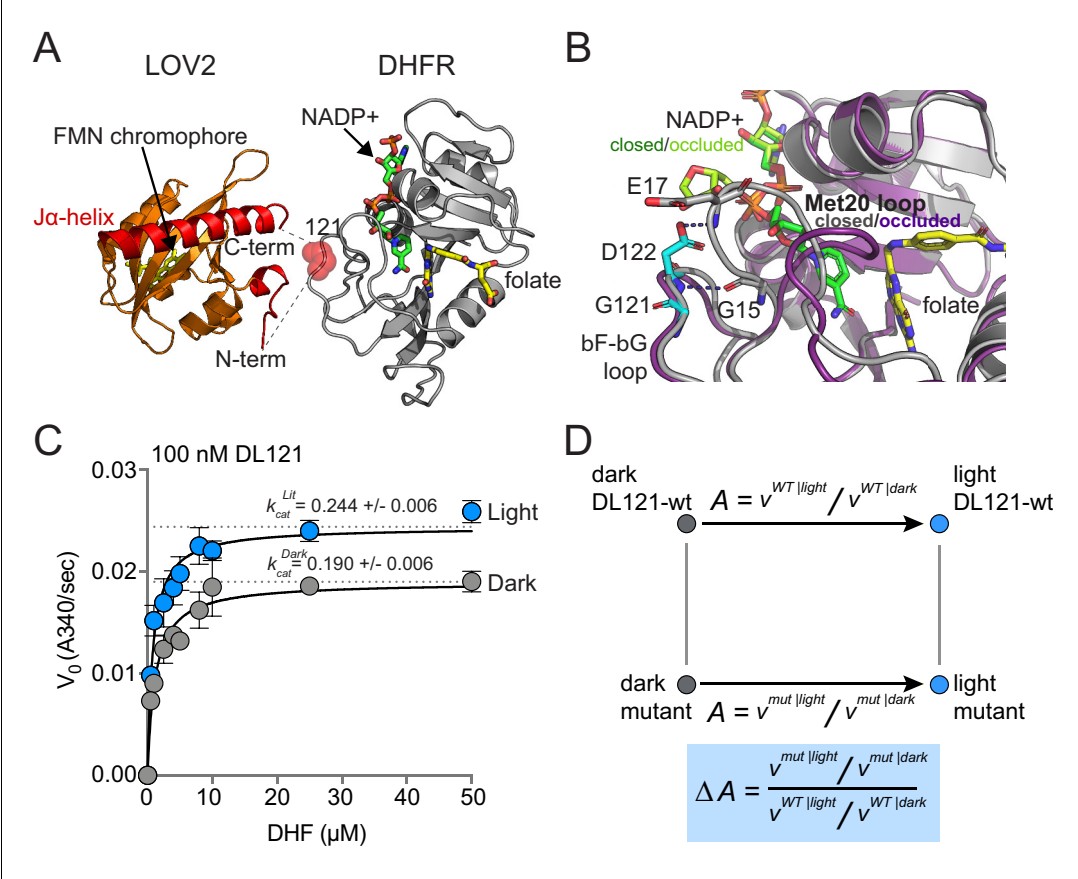

**Figure 1.** The DL121 DHFR/LOV2 fusion. (A) Composite structures of the individual DHFR and LOV2 domains (PDB ID: 1R × 2 and 2V0U), indicating the LOV2 insertion site between positions 120 and 121 of DHFR (*Sawaya and Kraut, 1997*; *Halavaty and Moffat, 2007*). DHFR is in gray cartoon, NADP co-factor in green sticks, and folate substrate in yellow sticks. In LOV2 signaling, blue light triggers the formation of a covalent adduct between a cysteine residue (C450) and a flavin mononucleotide (FMN, yellow sticks) (*Salomon et al., 2001*; *Crosson and Moffat, 2002*; *Swartz et al., 2002*) and associated unfolding of the C-terminal Jα-helix (red cartoon); this order-to-disorder transition is used for regulation in several synthetic and natural systems (*Pudasaini et al., 2015*; *Glantz et al., 2016*). (B) DHFR loop conformational changes near the LOV2 insertion site. While the mechanism of DHFR regulation by LOV2 is currently unknown, inspecting the native DHFR structure provides some insight. The substrate-bound Michaelis complex of native DHFR is in the 'closed' conformation (gray cartoon), while the product ternary complex is in the 'occluded' state (purple cartoon). The βF-βG loop, where LOV2 is inserted, is highlighted in cyan. In native DHFR, hydrogen bonds between this loop (Asp122) and the Met20 loop (Gly15, Glu17) are thought to stabilize the closed conformation (*Sawaya and Kraut, 1997*; *Schnell et al., 2004*). Mutations to positions 121 and 122 reduce activity and cause the enzyme to prefer the occluded conformation (*Cameron and Benkovic, 1997*; *Mhashal et al., 2018*; *Miller and Benkovic, 1998*). (C) Steady state Michaelis Menten kinetics for the DL121 fusion under lit (blue) and dark (gray) conditions. The $k_{cat}$ of DHFR increases 28% in response to light; the difference in $K_m$ is statistically insignificant (*Supplementary file 1a*). Error bars represent standard deviation for three replicates. (D) Quantifying the allosteric effect of mutation. Allostery for the DL121 fusion is reported as the ratio between lit and dark velocity. The effect of a mutation on allostery is then computed as the ratio of mutant allostery to wt-DL121 allostery (bottom blue box).

As in prior work, we measured the growth rate of many *E. coli* strains in parallel by using next generation sequencing (NGS) to monitor the frequency of individual DHFR mutants over time in a mixed culture (*Figure 2*; *Reynolds et al., 2011*; *Thompson et al., 2020*). Allele frequencies ($f_a$) at each time point (t) were normalized as follows: $f_a = \ln\left(\frac{N_a}{N_{WT}}\right)_t - \ln\left(\frac{N_a}{N_{WT}}\right)_{t=0}$ where $N_a$ and $N_{WT}$ are the number of mutant and wildtype (WT) counts at a given time point. By performing a linear fit of the log normalized allele frequencies vs. time we calculated a slope corresponding to relative growth rate: this value is the difference in growth rate for the mutant relative to a reference ('WT') construct.

As individual mutations tend to exhibit modest effects on allosteric regulation, we optimized the linear regime and resolution of the growth rate assay in two ways (*Reynolds et al., 2011*). First, we grew the *E. coli* populations in a turbidostat outfitted with blue LEDs to activate LOV2 (*Figure 2A*).

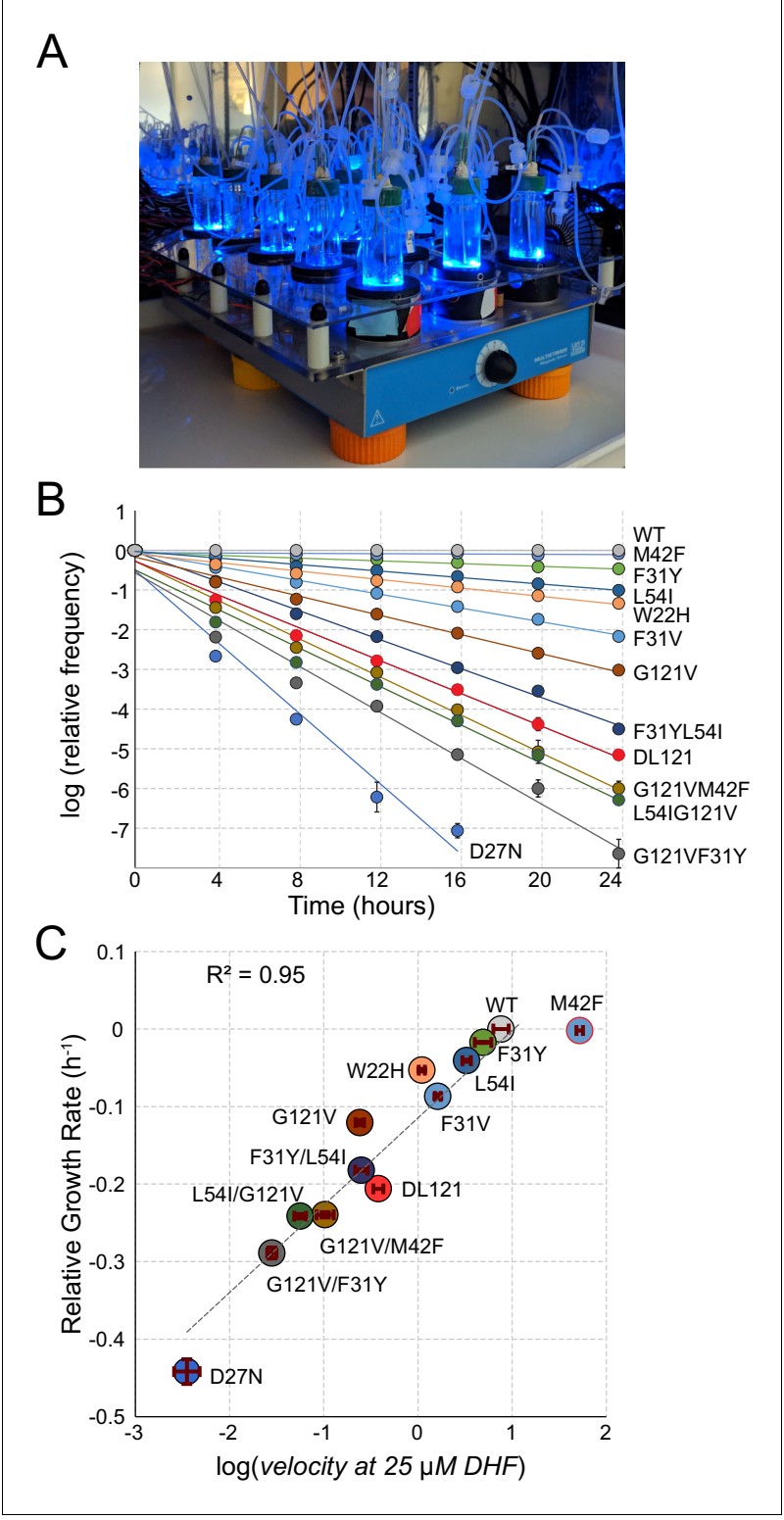

**Figure 2.** A high-throughput, high-resolution assay for DHFR activity. (**A**) The turbidostat. The instrument has 15 individual growth chambers (vials), positioned on a stir plate inside an incubator. Illumination was provided by blue LEDs in each vial holder. (**B**) Log-normalized relative allele frequency over time for 11 DHFR point mutations of known catalytic activity and the DL121 fusion. Allele frequency (colored circles) was determined by next-generation sequencing of mixed-population culture samples at each time point. All frequencies were normalized to t = 0 and WT DHFR (no LOV2 insertion). Error bars reflect standard error across four measurements, they are sometimes

*Figure 2 continued on next page*

*Figure 2 continued*

obscured by the marker. The slope for each line of best fit provides the growth rate of each mutant allele relative to WT DHFR. (C) Relative growth rate vs. log$_{10}$(velocity) for the 11 DHFR mutants and DL121 as characterized in panel B. Color coding of mutations is matched to panel B. Error bars reflect standard error of the mean over four replicates. The dashed line was fit by linear regression to all mutants in the linear regime (M42F excluded).

The turbidostat maintains each culture in exponential growth by dynamically sensing optical density and adjusting media dilution rate accordingly *Toprak et al., 2013*; this ensures near-constant media conditions and eliminates the need for manual serial dilutions. Second, we selected media conditions – M9 minimal media with 0.4% glucose and 1 µg/ml thymidine supplementation – in which growth rate can resolve subtle differences in catalytic activity near the DL121 fusion. We evaluated the resolution of our assay using a 'standard curve' of 11 point mutations of known catalytic activity in non-chimeric DHFR (*Figure 2B*). Under these conditions, we observed a log-linear relationship between relative growth rate and DHFR velocity over nearly four orders of magnitude; this relationship saturates (plateaus) for the most active mutants (WT and M42F, *Figure 2C*). Importantly, the relative growth rate and velocity of DL121 were near the center of the linear regime of our assay.

In using velocity to describe our data, we have incorporated two assumptions: (1) we presume minimal variation in protein abundance between mutants (enzyme concentration is equal to one) and (2) we fix the substrate concentration at 25 µM, which was previously reported as the endogenous concentration for WT *E. coli* (*Kwon et al., 2008*). Individual mutations may cause variation in protein abundance, but because allostery concerns a relative change in activity, light-independent differences in abundance can be removed by appropriate normalization (as discussed further below).

As previously observed, the exponential divergence of mutants with different growth rates in a population makes it possible to detect even small biochemical effects (*Breslow et al., 2008*). More specifically, we can discriminate a change of ±0.02 µM$^{-1}$ s$^{-1}$ in catalytic power ($k_{cat}/K_m$) under these conditions. This level of precision is on par with – and in some cases better than – literature-reported errors for in vitro steady state kinetics measurements of DHFR (*Reynolds et al., 2011*; *Wagner et al., 1992*; *Huang et al., 1994*). Consequently, we can resolve small catalytic and allosteric effects of mutations on DL121 through this high-throughput growth-based assay.

## Deleterious mutations are enriched at conserved, coevolving positions in DHFR

In order to map the coupling of individual DHFR positions to light, we constructed a deep mutational scanning library over all DHFR positions in the DL121 fusion (*Figure 3—figure supplements 1–2*). Then, we measured the growth rate effect of each mutation in triplicate under both lit and dark conditions using the above-described assay (*Figure 3A–C*, *Figure 3—figure supplements 3–4*, *Figure 3—source data 1*). In this experiment, all growth rates were calculated relative to the unmutated DL121 fusion, which itself exhibits reduced activity (and growth rate) compared to WT DHFR. Mutations fell into four broad categories in terms of growth rate effects: neutral, uniformly deleterious (*Figure 3A*), uniformly beneficial (*Figure 3B*), or light dependent (and thus allosteric, *Figure 3C*). We were unable to measure growth rate for 891 of the 3021 possible missense mutations (19 substitutions over 159 positions): 226 (7.5%) were missing at the start of the experiment (t = 0) for one or more replicates (referred to as 'no data'), and an additional 665 (22%) were depleted from the library before reaching the minimum of three time points required for growth rate estimation (we refer to these as null mutants, see also Materials and methods, *Figure 3—figure supplement 4*). We interpreted these 665 rapidly depleting null mutants as highly deleterious to growth rate and thus DHFR activity. The relative growth rates for the remaining 2130 mutations (70.5%) were highly reproducible, with a correlation coefficient between replicate pairs above 0.9 (*Figure 3—figure supplement 3*).

Before examining the allosteric effects of mutations, we first considered the effects of mutations on growth rate (and thus DHFR activity) in a single growth condition (dark). Prior work has found that deleterious mutations are enriched at evolutionarily conserved positions and within the protein sector (*McLaughlin et al., 2012*). The DHFR sector was defined by analyzing coevolution in a multiple sequence alignment of native DHFR domains, so we wished to examine if sector positions were indeed critical to function in the chimeric DL121 fusion. Good correspondence between the DHFR

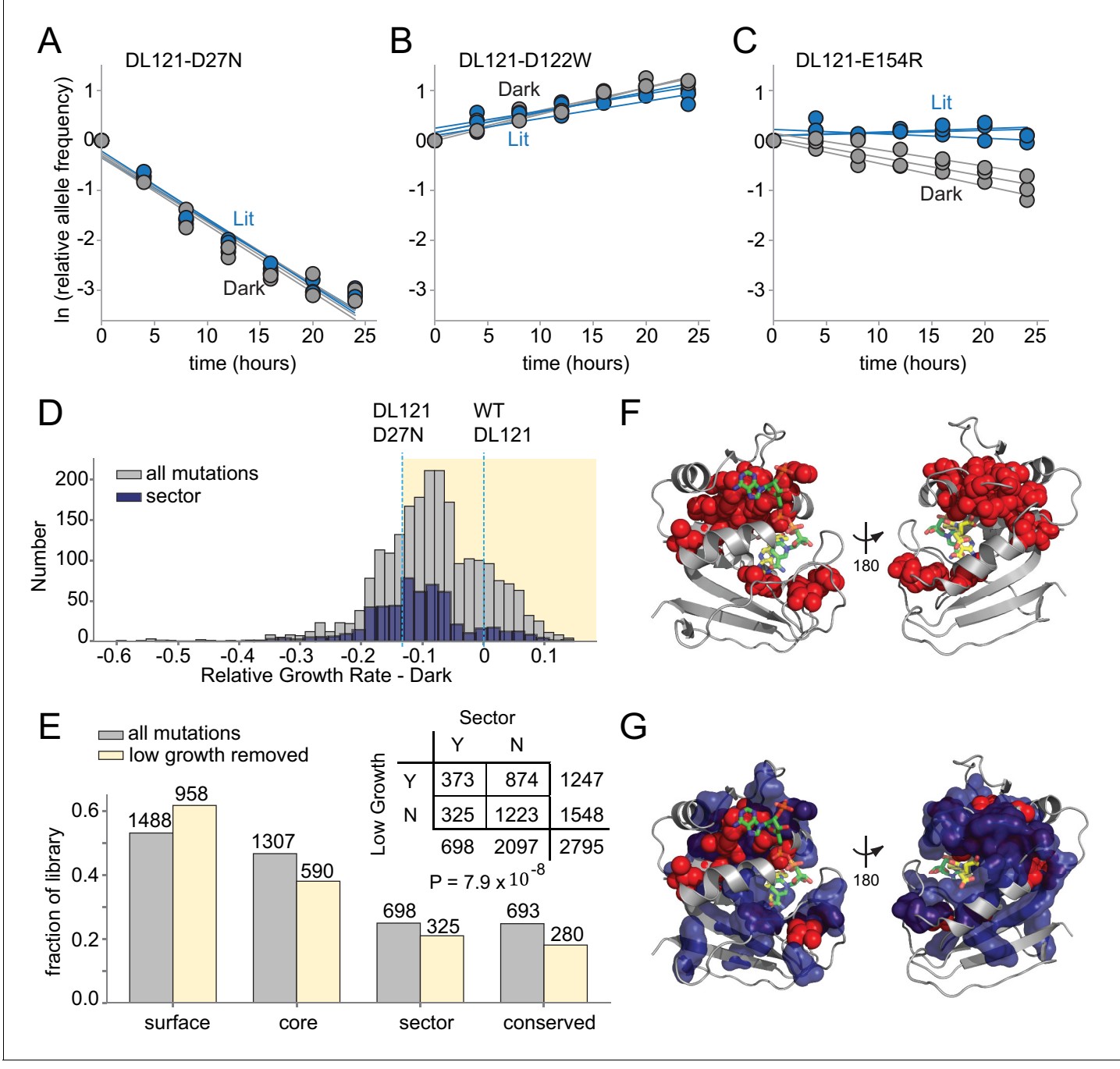

**Figure 3.** The effect of DL121 DHFR mutations on growth rate. (A–C) Representative relative growth rate trajectories for three mutations. (A) DL121 D27N was deleterious in both lit and dark conditions. (B) DL121 D122W was advantageous under both lit and dark conditions. (C) DL121 E154R was deleterious in the dark, and near neutral in the light. Solid lines were obtained by linear regression; the slope of these provides the difference in growth rate relative to the unmutated DL121 construct. Relative growth rates were measured in triplicate for each mutant under lit (blue) and dark (gray) conditions. (D) Distribution of relative growth rates under dark conditions. The distribution for all mutations with measurable growth rate effects is in gray ('null data' and 'no data' excluded); the distribution for sector mutations is in navy. The relative growth rate of DL121 D27N, a mutation that severely disrupts catalytic activity, is indicated with a cyan dashed line. (E) The fraction of DL121 mutations with measurable growth rates that can be categorized as: DHFR surface, core, sector, and evolutionarily conserved (see Materials and methods for definitions). The fraction is shown for both the complete library (gray bars), and the library after removing mutations with low growth (growth rate <= DL121 D27N). The absolute number of mutations is shown above each bar. A contingency table summarizes the overlap between mutations in the sector (at a p-value cutoff of 0.010), and the mutations that yield low growth (growth rate <= DL121 D27N). (F) Structural distribution of positions enriched for mutations with growth rates as low as or lower

*Figure 3 continued on next page*

*Figure 3 continued*

than DL121 D27N (red spheres). The DHFR backbone is in gray cartoon, the folate substrate in yellow sticks, and the NADP co-factor in green sticks. (**G**) Relationship of the sector (navy blue surface) to positions enriched for growth-rate disrupting mutations (red spheres, same as in F).

The online version of this article includes the following source data and figure supplement(s) for figure 3:

**Source data 1.** Relative growth rates under lit and dark conditions for DL121 point mutations as determined by next-generation sequencing.
**Figure supplement 1.** Deep mutational scanning library completeness – heatmap of counts for all mutants.
**Figure supplement 2.** Deep mutational scanning library completeness – distribution of counts for all mutants.
**Figure supplement 3.** Reproducibility across biological replicates.
**Figure supplement 4.** Heatmaps of relative growth rates.
**Figure supplement 5.** Growth rate measurements for DL121-D27N.
**Figure supplement 6.** Relationship between catalytically inactivating mutations and evolutionarily conserved positions.

sector, evolutionary conservation, and deleterious mutations in DL121 would provide confidence that the core functional elements of native DHFR remain intact in the chimera. The vast majority of mutations were at least modestly deleterious to growth, with a median relative growth rate of $-0.084$ in the dark and $-0.083$ in the light (*Figure 3D*). A cluster of beneficial mutations was observed just before the LOV2 insertion site at position 121 in both conditions, suggesting some potential to compensate for the inserted LOV2 (*Figure 3—figure supplement 4*). The overall distribution of fitness effects shows some differences relative to prior DMS studies of natural proteins including native *E. coli* DHFR (*Thompson et al., 2020*; *Garst et al., 2017*). First, the distribution of fitness effects for mutations in natural proteins is often centered around neutral, implying a certain degree of mutational robustness (*McLaughlin et al., 2012*; *Stiffler et al., 2015*). Secondly, DMS of native DHFR – under experimental conditions designed to resolve mutational effects near WT – revealed many beneficial (activating) mutations (*Thompson et al., 2020*). There are two explanations for the relative paucity of beneficial and neutral mutations in the present dataset. First, the DL121 fusion is comparably less robust because the unoptimized LOV2 insertion introduces an initial compromise to DHFR function. Secondly, the conditions of our assay (both expression and media) differ from prior work (*Thompson et al., 2020*) and were selected to resolve mutational effects near DL121; consequently, mutations with native-like (or better) activity are in the saturating, non-linear regime of our assay.

To identify the slowest growing – and presumably near, or entirely, inactivating – mutations, we applied an empirical growth rate cutoff of $-0.13$ to the lit and dark growth rates. This corresponds to the growth rate for DL121 D27N; D27N is an active site mutation that strongly reduces the activity of WT DHFR (*Figure 2B,C*). The DL121 D27N mutant grows very slowly in the conditions of our assay and is inviable in the absence of thymidine supplementation (*Figure 3—figure supplement 5*). We found that mutations with growth rates at or below this cutoff (including the null mutants) were significantly enriched in both the sector (p=$7.9\times10^{-8}$, *Figure 3E*, *Supplementary file 1b*) and at evolutionarily conserved positions (p=$8.7\times10^{-20}$, *Figure 3—figure supplement 6*, *Supplementary file 1c*). When mapped to the WT DHFR structure, positions enriched for deleterious mutations surround the active site and co-factor binding pocket (*Figure 3F*), structurally overlap with the sector (*Figure 3G*), and include a number of positions known to play a critical role in WT DHFR catalysis (e.g. W22, D27, M42, and L54) (*Howell et al., 1986*; *Fierke et al., 1987*). These data are consistent with the view that sector positions continue to play a key role in conferring DHFR catalytic activity in the DL121 fusion.

Following the thinking that (near) inactive DHFR variants are both inherently non-allosteric and associated with the least reproducible growth rate measurements (*Figure 3—figure supplement 3*), we removed the set of 1247 slow-growing (growth rate $<-0.13$) and null mutations prior to the analysis of allostery. The retained 1548 mutations – representing 51% of the growth assay data – remain well-distributed between the DL121 surface, core, sector, and evolutionarily conserved positions (*Figure 3E*). These present a high-confidence and representative subset of the data for evaluating mutational effects on DL121 allosteric regulation.

## Allostery tuning mutations are sparse

To compute the allosteric effect of mutation, we considered the triplicate measurements of lit and dark relative growth rate for each mutant (*Figure 3A–C*). Given the log-linear relationship between

growth rate and DHFR velocity (*Figure 2C*), subtracting growth rates approximates log-ratios of velocities. Thus, we estimated the allosteric effect of mutation by taking the difference in the average relative growth rates between lit and dark conditions:

In the above equations, rgr is relative growth rate (which is directly measured in our sequencing-based assay) and gr refers to absolute growth rate. Accordingly, positive values indicate allostery enhancing mutations and negative values indicate allostery disrupting mutations (*Figures 1D* and *4A*). Of the 1548 mutations evaluated, the allosteric effect is normally distributed with a mean near zero ($\mu = 0.0017$, *Figure 4—figure supplement 1*). To assess the statistical significance of allosteric effects, we computed a p-value for each mutation by unequal variance t-test under the null hypothesis that the lit and dark replicate measurements have equal means. These p-values were compared to a multiple-hypothesis testing adjusted p-value of p=0.016 determined by Sequential Goodness of Fit (SGoF, *Figure 4B*; *Carvajal-Rodriguez and de Uña-Alvarez, 2011*). Under these criteria, only 69 mutations (4.5% of all viable mutants) significantly influenced allostery: 56 mutations enhanced allostery while 13 disrupted allostery. We did not observe a strong association between the magnitude of growth rate effect and the allosteric effect size. Allostery-influencing mutations spanned a wide range of growth rates and exhibited comparatively modest effects on light regulation (*Figure 4C*).

To further examine the ability of the growth-based sequencing assay to quantitatively resolve mutation-associated changes in allosteric regulation, we selected 10 mutations spanning a range of allosteric and growth rate effects for in vitro characterization (*Figure 4B* red dots, *Figure 4—figure supplements 2–4*). As a control, we included the light insensitive variant DL121-C450S: the C450S mutation of LOV2 abrogates light-based signaling by blocking formation of a light-induced covalent bond between position 450 and the FMN chromophore (*Christie et al., 2002*). We expressed and purified the selected DL121 mutants to near homogeneity; S148C and E154R did not yield sufficient quantities of active protein for in vitro studies. We find it noteworthy that E154R—one of the strongest allostery-enhancing mutations in vivo—was unstable in multiple purification strategies. For the remaining eight mutations we measured the $k_{cat}$ and $K_m$ of DHFR under lit and dark conditions (*Figure 4—figure supplement 2*). To confirm function of the fused LOV2 domain, we also measured relaxation of the FMN chromophore following light stimulation and collected absorbance spectra before and after the application of light (*Figure 4—figure supplements 3–4*). As expected, all the characterized DL121 mutations (with the exception of DL121-C450S) retained LOV2 domains with light-responsive absorbance spectra and chromophore relaxation constants similar to the unmutated DL121 construct. Evaluating the light dependence of DHFR activity, the change in $K_m$ value between lit and dark conditions was neither significant for any point mutation nor correlated to allosteric effect size ($R^2 = 0.003$) (*Supplementary file 1a*, *Figure 4—figure supplements 5–6*). The $K_m$ values for all characterized mutants (0.15–1.9 μM) were similar to that of unmutated DL121 (~1 μM). Instead, we observed that light predominantly modulated catalytic turnover ($k_{cat}$). The ratio of $k_{cat}$ in the light relative to the dark ranged from 1.1 (for the non-allosteric DL121-C450S construct) to 2.0 (for the most allosteric point mutation, H124Q) (*Supplementary file 1a*, *Figure 4—figure supplements 5–6*). For reference, the starting DL121 construct has a lit:dark $k_{cat}$ ratio of 1.3. So why might the characterized allosteric mutations predominantly effect $k_{cat}$? One plausible explanation is that the conditions of our in vivo experiments fall within a pseudo-zero-order kinetics regime ([DHF] $\gg K_m$). In this scenario, light-associated changes in $K_m$ would have little impact on enzyme velocity (and accordingly growth rate) and go undetected in our assay. Consistent with this, the in vivo concentration of DHF for wildtype *E. coli* (25 μM) is well above the $K_m$ for all the characterized DL121 mutations. Alternatively, it could be that the biophysical mechanism of the DL121 fusion somehow makes it more energetically feasible for light to modulate $k_{cat}$ than $K_m$. In any case, the 1.3- to 2-fold changes in $k_{cat}$ translate to similar fold changes in enzyme velocity. A comparison of the in vitro allosteric effect on velocity to the in vivo growth rate effect yields a near-linear relationship with a correlation coefficient of 0.83 (*Figure 4D*). Taken together, these data show that our growth-based assay is quantitatively reporting on changes in allostery, and that the allosteric mutations identified here modulate DHFR activity through changes in catalytic turnover number.

## The structural pattern of allostery tuning mutations

Next, we examined the distribution of allostery-tuning mutations on the WT DHFR tertiary structure. The 13 allostery disrupting mutations localized to six DHFR positions concentrated near the LOV2 insertion site (*Figure 5A*). More specifically, 90% of the allostery disrupting mutations occurred

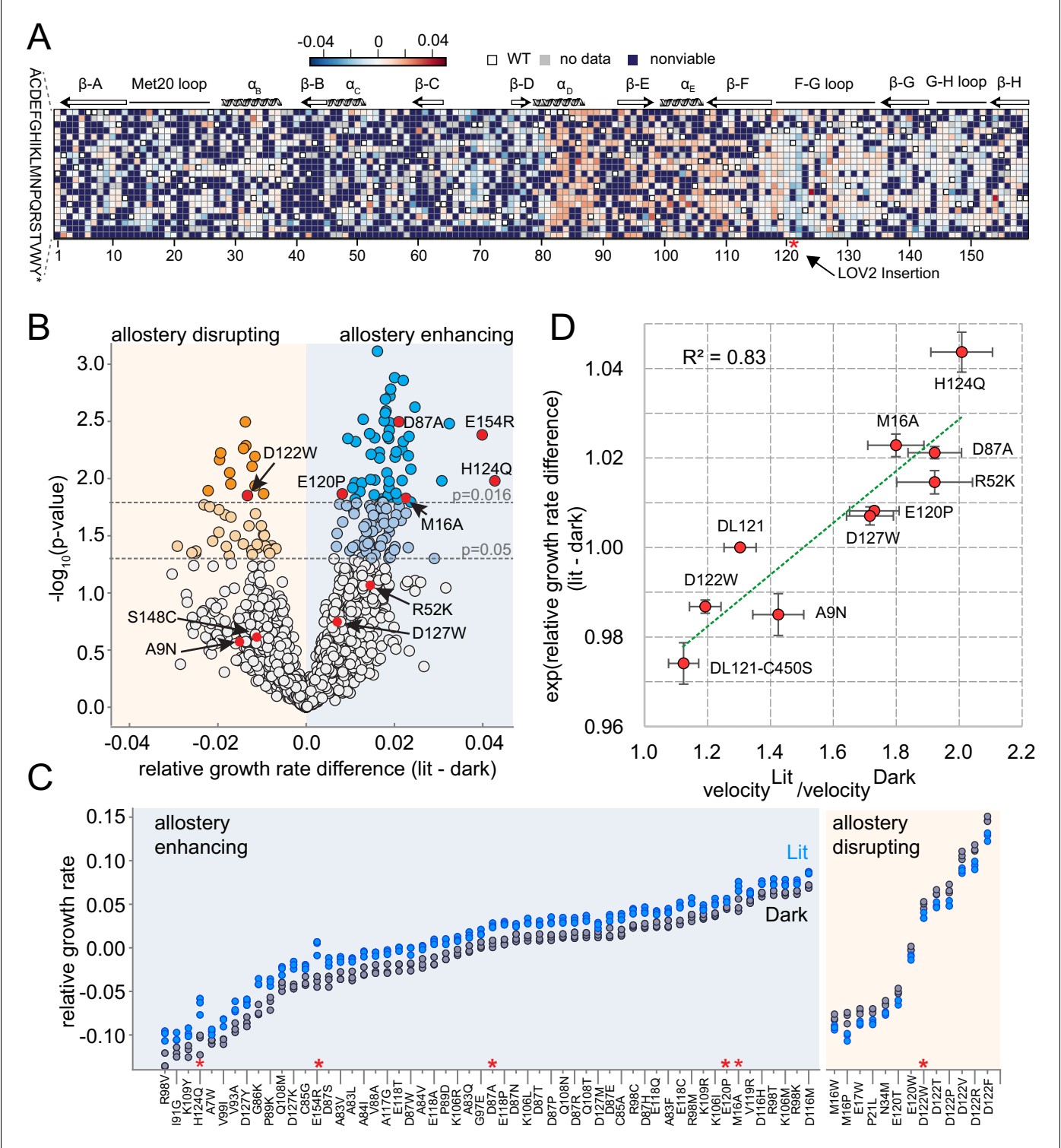

**Figure 4.** The effect of DL121 DHFR mutations on allostery. (**A**) Heatmap of mutational effects on allostery. Blue indicates allostery disrupting mutations, and red indicates allostery enhancing mutations. White squares with black outlines mark the WT residue at each position. Mutations missing from the library ('no data') are colored gray, and mutations that did not have sufficient sequencing counts for at least three time points ('null data') are colored navy. The LOV2 domain insertion site is indicated with a red star. (**B**) Volcano plot indicating the statistical significance of the light-dark growth rate difference (y-axis) as a function of relative growth rate difference (x-axis). p-Values were computed using a t-test across triplicate light and dark measurements. Individual points correspond to mutations; mutations on the left (yellow) side of the graph are allostery disrupting, while mutations on the right (blue) are allostery enhancing. Two cutoffs for statistical significance are indicated with dashed gray lines – both a standard value of p=0.05,

*Figure 4 continued on next page*

*Figure 4 continued*

and an adjusted p-value of 0.016, obtained by using Sequential Goodness of Fit (SGoF) to account for multiple hypothesis testing. Mutations selected for further in vitro experimental characterization are colored red and labeled. S148C and E154R did not yield sufficient quantities of active protein for further in vitro characterization. (**C**) Triplicate relative growth rate measurements under lit (blue) and dark (gray) conditions for all mutations with statistically significant allostery at the adjusted p-value (p<=0.016). The mutations are sorted by dark growth rate; mutations selected for in vitro characterization are marked with red asterisks. (**D**) Relationship between the allosteric effect as measured in vivo and in vitro. As we expect a log-linear relationship, we compare the ratio of velocity at 25 µM DHF (along x) to the exponent of the relative growth rate difference (along y). The relative growth rate difference under lit and dark conditions is the mean of triplicate measurements, error bars indicate SEM. All mutant effects on growth rate were measured in the same experiment (corresponding to a subset of the data in panel B) with the exception of DL121 C450S. The relative growth rate for this light-insensitive LOV2 mutant was measured in the 'calibration curve' experiment shown in *Figure 2* (see also Materials and methods). The ratio between velocity in the light and velocity in the dark reflects the mean of triplicate measurements; error bars indicate SEM. The green line was fit by linear regression.

The online version of this article includes the following figure supplement(s) for figure 4:

**Figure supplement 1.** Distribution of mutational effects on allosteric regulation.

**Figure supplement 2.** Steady state kinetics measurements for select mutants in the light and dark.

**Figure supplement 3.** Spectroscopic characterization of LOV2 activation for select DL121 mutants.

**Figure supplement 4.** Relaxation rate of the LOV2 chromophore for select DL121 mutants.

**Figure supplement 5.** Steady state kinetics parameters under lit and dark conditions for select mutants of the DL121 fusion.

**Figure supplement 6.** Correlation between in vivo allostery and in vitro steady state kinetics parameters for mutants of the DL121 fusion.

within 10 Å of the DHFR 121 cα atom (*Figure 5B*). These mutations were modestly enriched in the protein sector (*Supplementary file 1d*). Overall, the observed spatial distribution suggests these mutations may disrupt allostery by altering local structural contacts needed to ensure communication between DHFR and LOV2.

In contrast to this localized pattern, the 56 allostery enhancing mutations were observed at 25 positions distributed across the DHFR structure (*Figure 5C*) and enriched on the protein surface (*Figure 5D*, *Supplementary file 1e*). These enhancing mutations were never found in the protein sector and were thus statistically significantly depleted from the protein sector (*Figure 5E,F*). This relationship – wherein allostery disrupting mutations were modestly enriched and allostery enhancing mutations were strongly depleted from the sector – also holds when defining the set of allosteric mutations at a relaxed cutoff of p=0.05 (*Supplementary file 1d*). Given the prior finding that *sector connected* surface sites were hotspots for introducing allostery in DHFR (*Reynolds et al., 2011*), we also examined the association between allostery-influencing mutations and two other groups of DHFR positions: (1) surface sites that are either within or contacting the sector and (2) surface sites that are only contacting the sector (but not within-sector). As for the analysis of sector positions only, we observed a statistically significant depletion of allostery enhancing mutations and enrichment of allostery disrupting mutations when considering the set of surface sites within or contacting the sector. This finding holds true over a range of significance thresholds for defining sector and allosteric mutations (*Supplementary file 1f*). When considering the set of positions that contact (but are not within) the sector, we did not observe a statistically significant association at nearly all cutoffs (*Supplementary file 1g*). Indeed, a number of allostery enhancing mutations do not contact the sector at all and occur in surface exposed loops (e.g. from residues 84 to 89, and from 116 to 119). So, counter to our expectations, the optimization of allostery did not occur at sector connected sites or even proximal to the LOV2 insertion site. Instead, structurally distributed and weakly conserved surface sites provided a basis for tuning and enhancing allosteric regulation regardless of sector connectivity.

Taken together, our data show that many distributed surface sites can make modest contributions to allosteric regulation. Can these mutants be combined to further improve allosteric dynamic range? To test this, we created two mutant constructs by combining the most potent allostery enhancing mutations as characterized in vitro: the double mutant DL121-M16A,H124Q, and the triple mutant DL121-M16A,D87A,H124Q (*Figure 6A*). For both constructs, we measured steady-state catalytic parameters (*Supplementary file 1a*) and verified LOV2 function through absorbance spectra and chromophore relaxation kinetics experiments (*Figure 6—figure supplement 1*). Interestingly, all three mutations exhibited near-log-additive improvements in allostery (*Figure 6B*). The DL121-M16A,H124Q fusion exhibits a 2.74 fold increase in velocity upon light activation while the

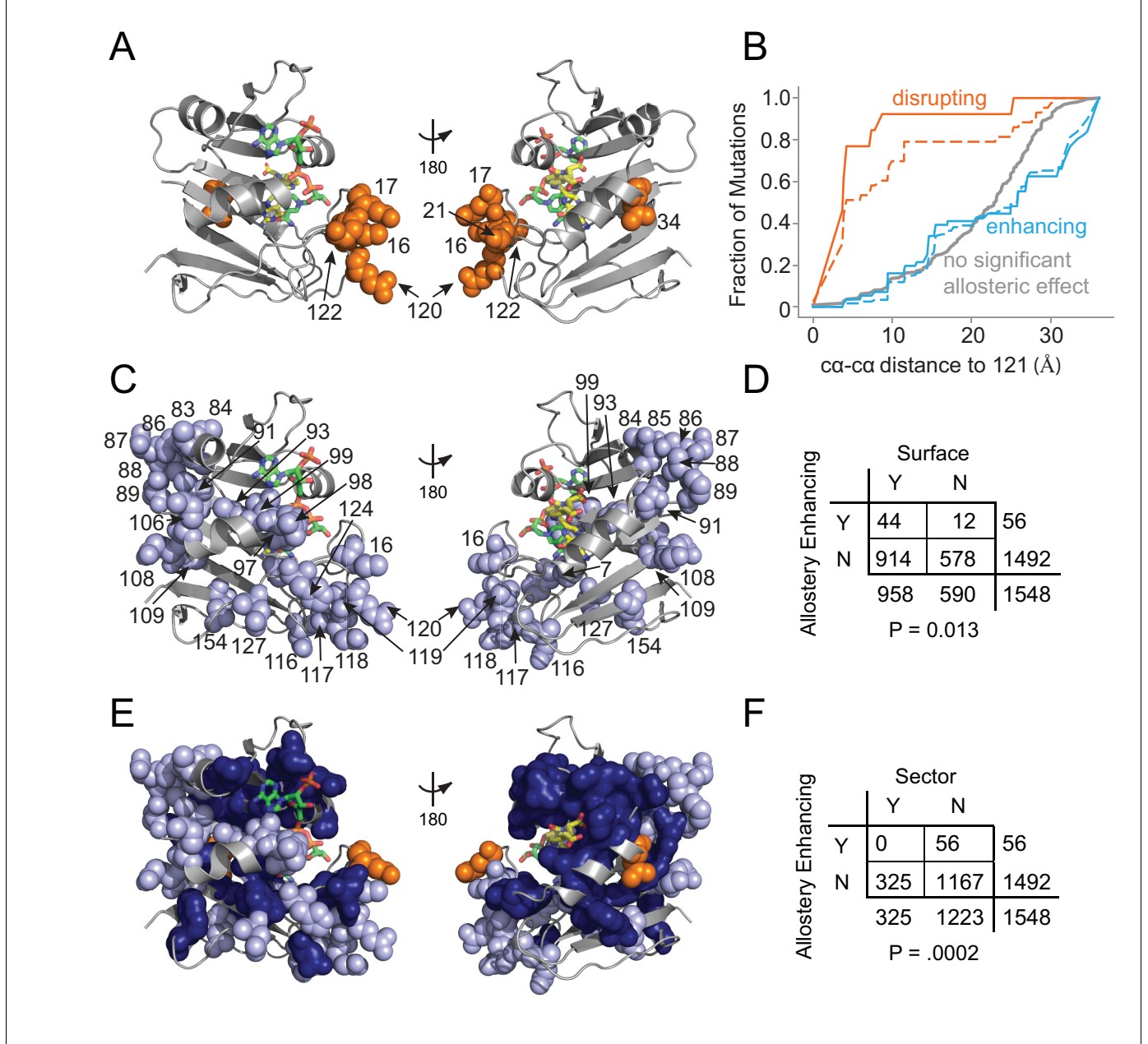

**Figure 5.** Structural distribution of allosteric mutations. (**A**) Sites of allostery disrupting mutations (orange spheres). DHFR backbone is in gray cartoon, folate substrate in yellow sticks, and NADP co-factor in green sticks. (**B**) Fraction of mutations that enhance (blue), disrupt (orange), or do not significantly influence allostery (gray) as a function of distance to the LOV2 insertion site at DHFR position 121. Solid and dashed lines indicate mutations at either the p=0.016 and p=0.05 significance cutoffs for allostery, respectively. (**C**) Sites of allostery enhancing mutations (light blue spheres). (**D**) Contingency table summarizing the overlap between allostery enhancing mutations and mutations on the DHFR solvent accessible surface (considered as >25% relative solvent accessibility in the 1R × 2 PDB). (**E**) Sites of allostery enhancing (light blue spheres) and disrupting mutations (orange spheres) in the context of the sector (dark blue surface). (**F**) Contingency table summarizing the relationship between allostery enhancing mutations and sector mutations (sector defined at a p-value cutoff of 0.010). No allostery enhancing mutations occur within the sector.

triple mutant shows a 3.87-fold increase in velocity. For both mutant combinations, the improvement in allostery is realized by reducing the dark state (constitutive) activity (*Figure 6—figure supplement 1*, *Supplementary file 1a*). The serial addition of allostery enhancing mutations also reduced the overall catalytic activity of DHFR, suggesting that further improvement could be obtained by combining these mutations with a non-allosteric but activity-enhancing mutation. Overall, these data suggest that a naïve sector connected fusion can be gradually evolved toward increased allosteric

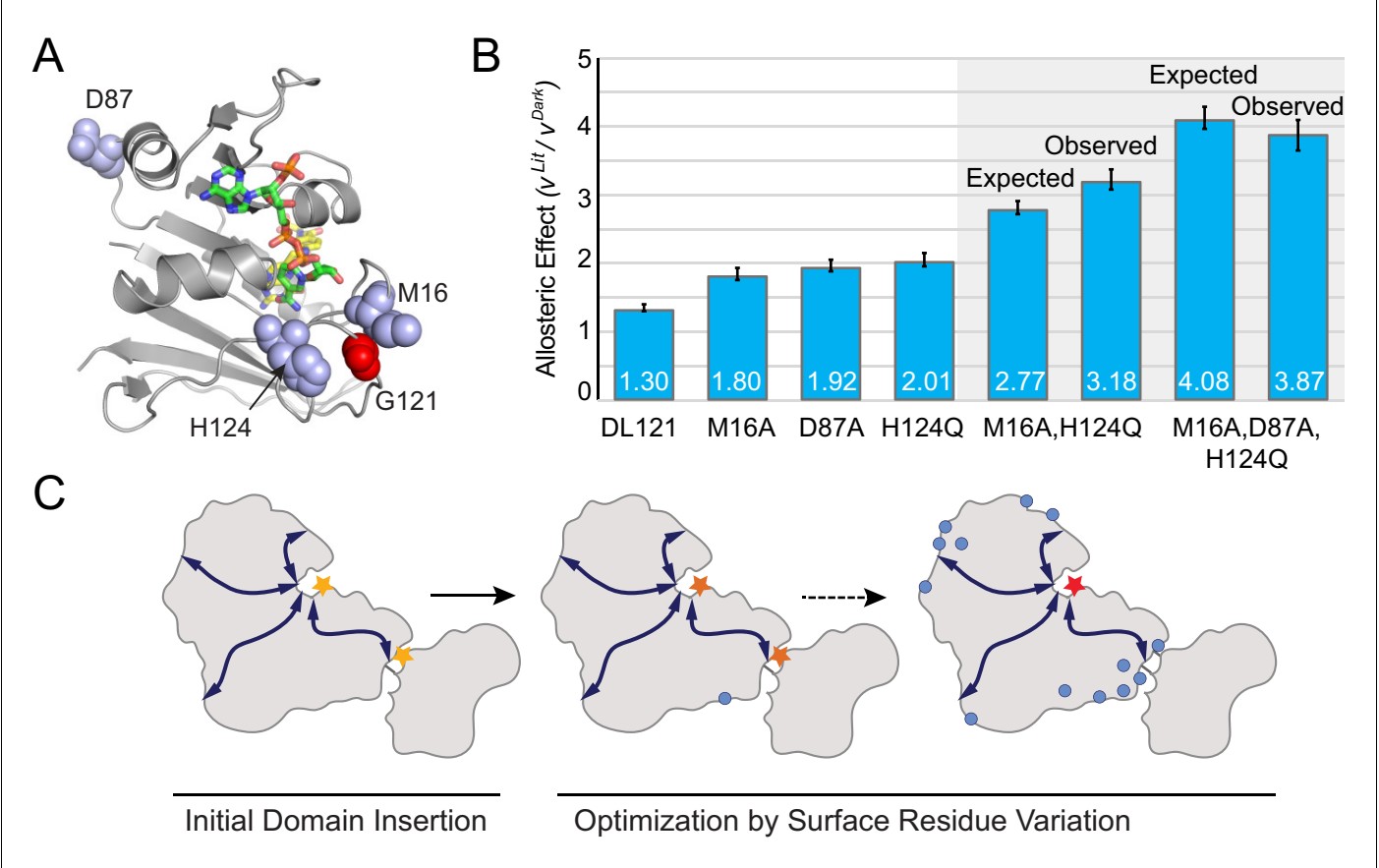

**Figure 6.** Combinatorial effect of allostery-enhancing mutations. (**A**) Location of M16, D87, and H124 (blue spheres). The LOV2 insertion site, G121, is shown in red spheres. The DHFR backbone is in gray cartoon, the folate substrate in yellow sticks, and the NADP co-factor in green sticks. (**B**) The in vitro allosteric effect of the single, double and triple mutants. Included are the log-additive expectations (Expected) for the double and triple mutants given only the single mutation effects, and the experimentally measured effects (Observed). The ratio between velocity in the light and dark reflects the mean of triplicate measurements; error bars indicate SEM. There is not a statistically significant difference between the expected and observed allosteric effects (p=0.07 for M16A,H124Q, p=0.48 for M16A,D87A,H124Q; as computed by unpaired t-test). (**C**) Schematic whereby a novel domain insertion is iteratively optimized by surface residue variation.

The online version of this article includes the following figure supplement(s) for figure 6:

**Figure supplement 1.** Characterization of the DL121- M16A,H124Q and DL121- M16A, D87A, H124Q mutants.

dynamic range through the stepwise accumulation of single mutations at structurally distributed surface sites (***Figure 6C***).

## Discussion

We used deep mutational scanning to study the frequency and structural pattern of allostery tuning mutations in a synthetic allosteric system, with the goal of understanding how regulation between domains can be optimized. Overall, allostery-influencing mutations were rare – just under 5% of viable mutations had statistically distinguishable effects on the lit and dark states of the DL121 fusion. We found that mutations at conserved and co-evolving (sector) positions were often deleterious to DHFR function and infrequently influenced allosteric regulation. In a few cases, sector mutations served to disrupt allostery; nearly all allostery disrupting mutations were localized to the LOV2 insertion site on DHFR. Counter to our expectations, allostery enhancing mutations were distributed across the DHFR structure, depleted from the sector, and enriched on the protein surface. When considered individually, the allostery-enhancing mutations had modest effects (up to twofold) on regulation, but (at least in some cases) they can be combined to yield near-additive improvements in

dynamic range. A triple mutant (DL121-M16A,D87A,H124Q) rationally designed using our point mutant data produces a 3.87-fold increase in velocity upon light stimulation, up from the 1.3-fold allosteric effect of our starting construct.

These results should be considered in the context of our experiment: the DL121 fusion begins with sharply reduced DHFR activity, and our experiment intentionally used relatively stringent DHFR selection conditions to better resolve small differences in kinetic parameters. Thus, it is unsurprising that a large fraction of DHFR mutations in our library were deleterious, with an appreciable fraction near-inactive. This result echoes prior studies showing that the fraction of deleterious mutations (and mutational robustness) is strongly modulated by a variety of factors, including purifying selection strength and expression level (*Stiffler et al., 2015*; *Jiang et al., 2013*; *Lundin et al., 2018*). Given the finding that stabilizing mutations can often improve protein evolvability (*Lundin et al., 2018*; *Bloom et al., 2006*; *Zheng et al., 2020*), it would be interesting to examine how the distribution of mutational effects on both DL121 function and allostery would change in the background of a stability (and/or activity) enhancing mutation to DL121. While we observed that the number of allosteric mutations is few and the effect sizes are generally small in our model system, a previous study of allostery tuning mutations in pyruvate kinase indicated that up to 30% of mutations can tune allostery, with the maximum observed effect size approaching 22-fold (*Tang and Fenton, 2017*). Nevertheless, our data serve to illuminate the pattern of mutational effects on a newly established (and unoptimized) domain fusion – the presumptive first step toward regulation in a number of both natural and synthetic systems.

Interestingly, we observe a seeming disparity between the sites where we were able to introduce new allosteric regulation by domain fusion (in our earlier work), and the sites where allosteric tuning takes place (in this work). Previously, Reynolds et al. found that sector connected surface sites served as hotspots for the introduction of new light-based regulation in DHFR (*Reynolds et al., 2011*). Indeed, allosteric regulation was never obtained when the LOV2 domain was inserted at a non-sector connected site. In contrast, in this work, we observed that allostery enhancing mutations were depleted both within the sector and at sector connected sites. For example, we observed a number of allostery enhancing mutations at positions 83–89 of the DHFR αD-βE loop, while LOV2 insertions in this region location did not initiate allostery as quantified either in vitro or in vivo (*Lee et al., 2008*; *Reynolds et al., 2011*). This suggests different structural requirements for establishing and tuning allostery in this system (and possibly others): here allostery seems to be more easily introduced at evolutionarily conserved and co-evolving sites, but once established, can be optimized through less conserved sector-peripheral residues.

Although our work focuses on a synthetic allosteric fusion, our results are broadly consistent with an emerging body of work characterizing allostery-influencing mutations in natural proteins. Together, these data point to a model in which mutations at evolutionarily conserved positions exert large (and often disruptive) effects on function while allostery is tuned at less conserved surface sites. For example, Leander et al. recently used deep mutational scanning to map the pattern of compensatory mutations that rescued allosteric function for non-allosteric tetracycline repressor (TetR) variants (*Leander et al., 2020*). In that study a 'disrupt-and-restore' strategy was used: an already-allosteric system was inactivated and deep mutational scanning was then used to identify compensatory mutations. While there are significant differences between rescuing a deficient variant and the optimization of a novel allosteric construct, they likewise found that the mutations at highly conserved sites were often disruptive to stability and function, while allostery-rescuing mutations occurred at weakly conserved and structurally distributed sites (*Leander et al., 2020*). Similarly, mutations at 'rheostat' sites – weakly conserved positions distal to the site of regulation – were found to modulate allosteric control in human liver pyruvate kinase and the lactose repressor protein (lacI) (*Campitelli et al., 2020*; *Wu et al., 2019*). Intriguingly, the association of allostery enhancing mutations with the protein surface hints at a possible role for solvent – and more specifically the protein hydration layer – in tuning regulation.

The finding that the allostery initiated upon naïve fusion of the DHFR and LOV2 domains can be further enhanced by single mutations implies a path to improved allosteric dynamic range by stepwise mutagenesis and selection. Three of the most allostery enhancing mutations could be combined to yield a near-additive improvement in regulatory dynamic range. This has interesting implications for both evolved and engineered allosteric systems. In evolved systems, standing mutational variation is more likely at weakly conserved surface sites (particularly under less stringent

selection conditions), and this could provide a means for generating variation in allosteric regulation upon a domain fusion event. Moreover, while engineering studies sometimes use mutations near the domain insertion site to optimize regulation, our results suggest that diffuse surface site mutations could present an effective alternative. Whether by engineering or evolution, it seems that mutations at weakly conserved and structurally distributed residues can provide a path to the optimization of regulation.

# Materials and methods

## Key resources table

| Reagent type (species) or resource | Designation | Source or reference | Identifiers | Additional information |
|---|---|---|---|---|
| Gene (*Escherichia coli*) | DHFR-LOV2 121 | Reynolds et al. Cell 2011 [20] | Fusion of *Escherichia coli* DHFR and *Avena sativa* LOV2 | |
| Strain, strain background (*Escherichia coli*) | BL21(DE3) | New England Biolabs | NEB #: C2527H | Competent cells |
| Strain, strain background (*Escherichia coli*) | ER2566 ΔfolA ΔthyA | Dr. Steven Benkovic, described in [20, 26] | Competent cells | |
| Strain, strain background (*Escherichia coli*) | XL1-Blue | Agilent Technologies | Cat. #: 200249 | Competent cells |
| Recombinant DNA reagent | pACYC-Duet_DL121_WTTS (plasmid) | Reynolds et al. Cell 2011 [20] | Addgene ID 171954 | Contains chimeric DL121 with TYMS (selection vector) |
| Recombinant DNA reagent | pHIS8-3_DL121 (plasmid) | Reynolds et al. Cell 2011 [20] | Addgene ID 171953 | Contains chimeric DL121 (expression vector) |
| Sequence-based reagent | DL121_pos1_fwd | This Paper | Mutagenic PCR primer | NNSATCAGTCTGATTGCGGCG |
| Sequence-based reagent | DL121_pos2_fwd | This Paper | Mutagenic PCR primer | NNSAGTCTGATTGCGGCGTTAG |
| Sequence-based reagent | DL121_pos3_fwd | This Paper | Mutagenic PCR primer | NNSCTGATTGCGGCGTTAGCG |
| Sequence-based reagent | DL121_pos4_fwd | This Paper | Mutagenic PCR primer | NNSATTGCGGCGTTAGCGGTA |
| Sequence-based reagent | DL121_pos5_fwd | This Paper | Mutagenic PCR primer | NNSGCGGCGTTAGCGGTAGAT |
| Sequence-based reagent | DL121_pos6_fwd | This Paper | Mutagenic PCR primer | NNSGCGTTAGCGGTAGATCGC |
| Sequence-based reagent | DL121_pos7_fwd | This Paper | Mutagenic PCR primer | NNSTTAGCGGTAGATCGCGTTATC |
| Sequence-based reagent | DL121_pos8_fwd | This Paper | Mutagenic PCR primer | NNSGCGGTAGATCGCGTTATCG |
| Sequence-based reagent | DL121_pos9_fwd | This Paper | Mutagenic PCR primer | NNSGTAGATCGCGTTATCGGCATG |
| Sequence-based reagent | DL121_pos10_fwd | This Paper | Mutagenic PCR primer | NNSGATCGCGTTATCGGCATGG |
| Sequence-based reagent | DL121_pos11_fwd | This Paper | Mutagenic PCR primer | NNSCGCGTTATCGGCATGGAAAA |
| Sequence-based reagent | DL121_pos12_fwd | This Paper | Mutagenic PCR primer | NNSGTTATCGGCATGGAAAACGC |
| Sequence-based reagent | DL121_pos13_fwd | This Paper | Mutagenic PCR primer | NNSATCGGCATGGAAAACGCC |
| Sequence-based reagent | DL121_pos14_fwd | This Paper | Mutagenic PCR primer | NNSGGCATGGAAAACGCCATG |

*Continued on next page*

*Continued*

| Reagent type (species) or resource | Designation | Source or reference | Identifiers | Additional information |
|---|---|---|---|---|
| Sequence-based reagent | DL121_pos15_fwd | This Paper | Mutagenic PCR primer | NNSATGGAAAACGCCATGCCG |
| Sequence-based reagent | DL121_pos16_fwd | This Paper | Mutagenic PCR primer | NNSGAAAACGCCATGCCGTGG |
| Sequence-based reagent | DL121_pos17_fwd | This Paper | Mutagenic PCR primer | NNSAACGCCATGCCGTGGAAC |
| Sequence-based reagent | DL121_pos18_fwd | This Paper | Mutagenic PCR primer | NNSGCCATGCCGTGGAACCTG |
| Sequence-based reagent | DL121_pos19_fwd | This Paper | Mutagenic PCR primer | NNSATGCCGTGGAACCTGCCT |
| Sequence-based reagent | DL121_pos20_fwd | This Paper | Mutagenic PCR primer | NNSCCGTGGAACCTGCCTGCC |
| Sequence-based reagent | DL121_pos21_fwd | This Paper | Mutagenic PCR primer | NNSTGGAACCTGCCTGCCGAT |
| Sequence-based reagent | DL121_pos22_fwd | This Paper | Mutagenic PCR primer | NNSAACCTGCCTGCCGATCTC |
| Sequence-based reagent | DL121_pos23_fwd | This Paper | Mutagenic PCR primer | NNSCTGCCTGCCGATCTCGCC |
| Sequence-based reagent | DL121_pos24_fwd | This Paper | Mutagenic PCR primer | NNSCCTGCCGATCTCGCCTGG |
| Sequence-based reagent | DL121_pos25_fwd | This Paper | Mutagenic PCR primer | NNSGCCGATCTCGCCTGGTTT |
| Sequence-based reagent | DL121_pos26_fwd | This Paper | Mutagenic PCR primer | NNSGATCTCGCCTGGTTTAAACGC |
| Sequence-based reagent | DL121_pos27_fwd | This Paper | Mutagenic PCR primer | NNSCTCGCCTGGTTTAAACGCAACA |
| Sequence-based reagent | DL121_pos28_fwd | This Paper | Mutagenic PCR primer | NNSGCCTGGTTTAAACGCAACAC |
| Sequence-based reagent | DL121_pos29_fwd | This Paper | Mutagenic PCR primer | NNSTGGTTTAAACGCAACACCTTAAATAAAC |
| Sequence-based reagent | DL121_pos30_fwd | This Paper | Mutagenic PCR primer | NNSTTTAAACGCAACACCTTAAATAAACCCG |
| Sequence-based reagent | DL121_pos31_fwd | This Paper | Mutagenic PCR primer | NNSAAACGCAACACCTTAAATAAACCCGTG |
| Sequence-based reagent | DL121_pos32_fwd | This Paper | Mutagenic PCR primer | NNSCGCAACACCTTAAATAAACCCGT |
| Sequence-based reagent | DL121_pos33_fwd | This Paper | Mutagenic PCR primer | NNSAACACCTTAAATAAACCCGTGATTATGG |
| Sequence-based reagent | DL121_pos34_fwd | This Paper | Mutagenic PCR primer | NNSACCTTAAATAAACCCGTGATTATGGG |
| Sequence-based reagent | DL121_pos35_fwd | This Paper | Mutagenic PCR primer | NNSTTAAATAAACCCGTGATTATGGGCC |
| Sequence-based reagent | DL121_pos36_fwd | This Paper | Mutagenic PCR primer | NNSAATAAACCCGTGATTATGGGCC |
| Sequence-based reagent | DL121_pos37_fwd | This Paper | Mutagenic PCR primer | NNSAAACCCGTGATTATGGGCC |
| Sequence-based reagent | DL121_pos38_fwd | This Paper | Mutagenic PCR primer | NNSCCCGTGATTATGGGCCGC |
| Sequence-based reagent | DL121_pos39_fwd | This Paper | Mutagenic PCR primer | NNSGTGATTATGGGCCGCCATAC |
| Sequence-based reagent | DL121_pos40_fwd | This Paper | Mutagenic PCR primer | NNSATTATGGGCCGCCATACCT |

*Continued on next page*

*Continued*

| Reagent type (species) or resource | Designation | Source or reference | Identifiers | Additional information |
|---|---|---|---|---|
| Sequence-based reagent | DL121_pos41_fwd | This Paper | Mutagenic PCR primer | NNSATGGGCCGCCATACCTGG |
| Sequence-based reagent | DL121_pos42_fwd | This Paper | Mutagenic PCR primer | NNSGGCCGCCATACCTGGGAA |
| Sequence-based reagent | DL121_pos43_fwd | This Paper | Mutagenic PCR primer | NNSCGCCATACCTGGGAATCG |
| Sequence-based reagent | DL121_pos44_fwd | This Paper | Mutagenic PCR primer | NNSCATACCTGGGAATCGATCGGT |
| Sequence-based reagent | DL121_pos45_fwd | This Paper | Mutagenic PCR primer | NNSACCTGGGAATCGATCGGT |
| Sequence-based reagent | DL121_pos46_fwd | This Paper | Mutagenic PCR primer | NNSTGGGAATCGATCGGTCGT |
| Sequence-based reagent | DL121_pos47_fwd | This Paper | Mutagenic PCR primer | NNSGAATCGATCGGTCGTCCG |
| Sequence-based reagent | DL121_pos48_fwd | This Paper | Mutagenic PCR primer | NNSTCGATCGGTCGTCCGTTG |
| Sequence-based reagent | DL121_pos49_fwd | This Paper | Mutagenic PCR primer | NNSATCGGTCGTCCGTTGCCA |
| Sequence-based reagent | DL121_pos50_fwd | This Paper | Mutagenic PCR primer | NNSGGTCGTCCGTTGCCAGGA |
| Sequence-based reagent | DL121_pos51_fwd | This Paper | Mutagenic PCR primer | NNSCGTCCGTTGCCAGGACGC |
| Sequence-based reagent | DL121_pos52_fwd | This Paper | Mutagenic PCR primer | NNSCCGTTGCCAGGACGCAAA |
| Sequence-based reagent | DL121_pos53_fwd | This Paper | Mutagenic PCR primer | NNSTTGCCAGGACGCAAAAATATTATCC |
| Sequence-based reagent | DL121_pos54_fwd | This Paper | Mutagenic PCR primer | NNSCCAGGACGCAAAAATATTATCCTGAG |
| Sequence-based reagent | DL121_pos55_fwd | This Paper | Mutagenic PCR primer | NNSGGACGCAAAAATATTATCCTGAGCTC |
| Sequence-based reagent | DL121_pos56_fwd | This Paper | Mutagenic PCR primer | NNSCGCAAAAATATTATCCTGAGCTCACAA |
| Sequence-based reagent | DL121_pos57_fwd | This Paper | Mutagenic PCR primer | NNSAAAAATATTATCCTGAGCTCACAACCGG |
| Sequence-based reagent | DL121_pos58_fwd | This Paper | Mutagenic PCR primer | NNSAATATTATCCTGAGCTCACAACCGGGTA |
| Sequence-based reagent | DL121_pos59_fwd | This Paper | Mutagenic PCR primer | NNSATTATCCTGAGCTCACAACCG |
| Sequence-based reagent | DL121_pos60_fwd | This Paper | Mutagenic PCR primer | NNSATCCTGAGCTCACAACCG |
| Sequence-based reagent | DL121_pos61_fwd | This Paper | Mutagenic PCR primer | NNSCTGAGCTCACAACCGGGT |
| Sequence-based reagent | DL121_pos62_fwd | This Paper | Mutagenic PCR primer | NNSAGCTCACAACCGGGTACG |
| Sequence-based reagent | DL121_pos63_fwd | This Paper | Mutagenic PCR primer | NNSTCACAACCGGGTACGGAC |
| Sequence-based reagent | DL121_pos64_fwd | This Paper | Mutagenic PCR primer | NNSCAACCGGGTACGGACGAT |
| Sequence-based reagent | DL121_pos65_fwd | This Paper | Mutagenic PCR primer | NNSCCGGGTACGGACGATCGC |
| Sequence-based reagent | DL121_pos66_fwd | This Paper | Mutagenic PCR primer | NNSGGTACGGACGATCGCGTA |

*Continued on next page*

*Continued*

| Reagent type (species) or resource | Designation | Source or reference | Identifiers | Additional information |
|---|---|---|---|---|
| Sequence-based reagent | DL121_pos67_fwd | This Paper | Mutagenic PCR primer | NNSACGGACGATCGCGTAACG |
| Sequence-based reagent | DL121_pos68_fwd | This Paper | Mutagenic PCR primer | NNSGACGATCGCGTAACGTGG |
| Sequence-based reagent | DL121_pos69_fwd | This Paper | Mutagenic PCR primer | NNSGATCGCGTAACGTGGGTG |
| Sequence-based reagent | DL121_pos70_fwd | This Paper | Mutagenic PCR primer | NNSCGCGTAACGTGGGTGAAG |
| Sequence-based reagent | DL121_pos71_fwd | This Paper | Mutagenic PCR primer | NNSGTAACGTGGGTGAAGTCGG |
| Sequence-based reagent | DL121_pos72_fwd | This Paper | Mutagenic PCR primer | NNSACGTGGGTGAAGTCGGTG |
| Sequence-based reagent | DL121_pos73_fwd | This Paper | Mutagenic PCR primer | NNSTGGGTGAAGTCGGTGGAT |
| Sequence-based reagent | DL121_pos74_fwd2 | This Paper | Mutagenic PCR primer | NNSGTGAAGTCGGTGGATGAAG |
| Sequence-based reagent | DL121_pos75_fwd | This Paper | Mutagenic PCR primer | NNSAAGTCGGTGGATGAAGCAATTG |
| Sequence-based reagent | DL121_pos76_fwd | This Paper | Mutagenic PCR primer | NNSTCGGTGGATGAAGCAATTGC |
| Sequence-based reagent | DL121_pos77_fwd | This Paper | Mutagenic PCR primer | NNSGTGGATGAAGCAATTGCGG |
| Sequence-based reagent | DL121_pos78_fwd | This Paper | Mutagenic PCR primer | NNSGATGAAGCAATTGCGGCG |
| Sequence-based reagent | DL121_pos79_fwd | This Paper | Mutagenic PCR primer | NNSGAAGCAATTGCGGCGTGT |
| Sequence-based reagent | DL121_pos80_fwd | This Paper | Mutagenic PCR primer | NNSGCAATTGCGGCGTGTGGT |
| Sequence-based reagent | DL121_pos81_fwd | This Paper | Mutagenic PCR primer | NNSATTGCGGCGTGTGGTGAC |
| Sequence-based reagent | DL121_pos82_fwd | This Paper | Mutagenic PCR primer | NNSGCGGCGTGTGGTGACGTAC |
| Sequence-based reagent | DL121_pos83_fwd | This Paper | Mutagenic PCR primer | NNSGCGTGTGGTGACGTACCA |
| Sequence-based reagent | DL121_pos84_fwd | This Paper | Mutagenic PCR primer | NNSTGTGGTGACGTACCAGAAATCAT |
| Sequence-based reagent | DL121_pos85_fwd | This Paper | Mutagenic PCR primer | NNSGGTGACGTACCAGAAATCATGG |
| Sequence-based reagent | DL121_pos86_fwd | This Paper | Mutagenic PCR primer | NNSGACGTACCAGAAATCATGGTGATTG |
| Sequence-based reagent | DL121_pos87_fwd | This Paper | Mutagenic PCR primer | NNSGTACCAGAAATCATGGTGATTGGC |
| Sequence-based reagent | DL121_pos88_fwd | This Paper | Mutagenic PCR primer | NNSCCAGAAATCATGGTGATTGGC |
| Sequence-based reagent | DL121_pos89_fwd | This Paper | Mutagenic PCR primer | NNSGAAATCATGGTGATTGGCGG |
| Sequence-based reagent | DL121_pos90_fwd | This Paper | Mutagenic PCR primer | NNSATCATGGTGATTGGCGGC |
| Sequence-based reagent | DL121_pos91_fwd | This Paper | Mutagenic PCR primer | NNSATGGTGATTGGCGGCGGC |
| Sequence-based reagent | DL121_pos92_fwd | This Paper | Mutagenic PCR primer | NNSGTGATTGGCGGCGGCCGC |

*Continued on next page*

*Continued*

| Reagent type (species) or resource | Designation | Source or reference | Identifiers | Additional information |
|---|---|---|---|---|
| Sequence-based reagent | DL121_pos93_fwd | This Paper | Mutagenic PCR primer | NNSATTGGCGGCGGCCGCGTT |
| Sequence-based reagent | DL121_pos94_fwd | This Paper | Mutagenic PCR primer | NNSGGCGGCGGCCGCGTTTAT |
| Sequence-based reagent | DL121_pos95_fwd | This Paper | Mutagenic PCR primer | NNSGGCGGCCGCGTTTATGAA |
| Sequence-based reagent | DL121_pos96_fwd | This Paper | Mutagenic PCR primer | NNSGGCCGCGTTTATGAACAGTT |
| Sequence-based reagent | DL121_pos97_fwd | This Paper | Mutagenic PCR primer | NNSCGCGTTTATGAACAGTTCTTGC |
| Sequence-based reagent | DL121_pos98_fwd | This Paper | Mutagenic PCR primer | NNSGTTTATGAACAGTTCTTGCCAAAAGCGC |
| Sequence-based reagent | DL121_pos99_fwd | This Paper | Mutagenic PCR primer | NNSTATGAACAGTTCTTGCCAAAAGCGCAAA |
| Sequence-based reagent | DL121_pos100_fwd | This Paper | Mutagenic PCR primer | NNSGAACAGTTCTTGCCAAAAGCGCAAAAGC |
| Sequence-based reagent | DL121_pos101_fwd | This Paper | Mutagenic PCR primer | NNSCAGTTCTTGCCAAAAGCGCAAAAGCTTT |
| Sequence-based reagent | DL121_pos102_fwd | This Paper | Mutagenic PCR primer | NNSTTCTTGCCAAAAGCGCAAAAG |
| Sequence-based reagent | DL121_pos103_fwd | This Paper | Mutagenic PCR primer | NNSTTGCCAAAAGCGCAAAAGC |
| Sequence-based reagent | DL121_pos104_fwd | This Paper | Mutagenic PCR primer | NNSCCAAAAGCGCAAAAGCTTTATCTG |
| Sequence-based reagent | DL121_pos105_fwd | This Paper | Mutagenic PCR primer | NNSAAAGCGCAAAAGCTTTATCTGACG |
| Sequence-based reagent | DL121_pos106_fwd | This Paper | Mutagenic PCR primer | NNSGCGCAAAAGCTTTATCTGACG |
| Sequence-based reagent | DL121_pos107_fwd | This Paper | Mutagenic PCR primer | NNSCAAAAGCTTTATCTGACGCATATCGAC |
| Sequence-based reagent | DL121_pos108_fwd | This Paper | Mutagenic PCR primer | NNSAAGCTTTATCTGACGCATATCGAC |
| Sequence-based reagent | DL121_pos109_fwd | This Paper | Mutagenic PCR primer | NNSCTTTATCTGACGCATATCGACGC |
| Sequence-based reagent | DL121_pos110_fwd | This Paper | Mutagenic PCR primer | NNSTATCTGACGCATATCGACGCA |
| Sequence-based reagent | DL121_pos111_fwd | This Paper | Mutagenic PCR primer | NNSCTGACGCATATCGACGCAG |
| Sequence-based reagent | DL121_pos112_fwd | This Paper | Mutagenic PCR primer | NNSACGCATATCGACGCAGAAGT |
| Sequence-based reagent | DL121_pos113_fwd | This Paper | Mutagenic PCR primer | NNSCATATCGACGCAGAAGTGGAAC |
| Sequence-based reagent | DL121_pos114_fwd | This Paper | Mutagenic PCR primer | NNSATCGACGCAGAAGTGGAACT |
| Sequence-based reagent | DL121_pos115_fwd | This Paper | Mutagenic PCR primer | NNSGACGCAGAAGTGGAACTGG |
| Sequence-based reagent | DL121_pos116_fwd | This Paper | Mutagenic PCR primer | NNSGCAGAAGTGGAACTGGCC |
| Sequence-based reagent | DL121_pos117_fwd | This Paper | Mutagenic PCR primer | NNSGAAGTGGAACTGGCCACC |
| Sequence-based reagent | DL121_pos118_fwd | This Paper | Mutagenic PCR primer | NNSGTGGAACTGGCCACCACT |

*Continued*

| Reagent type (species) or resource | Designation | Source or reference | Identifiers | Additional information |
|---|---|---|---|---|
| Sequence-based reagent | DL121_pos119_fwd | This Paper | Mutagenic PCR primer | NNSGAACTGGCCACCACTCTAGA |
| Sequence-based reagent | DL121_pos120_fwd | This Paper | Mutagenic PCR primer | NNSCTGGCCACCACTCTAGAG |
| Sequence-based reagent | DL121_pos121_fwd | This Paper | Mutagenic PCR primer | NNSGACACCCATTTCCCGGATTAC |
| Sequence-based reagent | DL121_pos122_fwd | This Paper | Mutagenic PCR primer | NNSACCCATTTCCCGGATTACGA |
| Sequence-based reagent | DL121_pos123_fwd | This Paper | Mutagenic PCR primer | NNSCATTTCCCGGATTACGAGCC |
| Sequence-based reagent | DL121_pos124_fwd | This Paper | Mutagenic PCR primer | NNSTTCCCGGATTACGAGCCG |
| Sequence-based reagent | DL121_pos125_fwd | This Paper | Mutagenic PCR primer | NNSCCGGATTACGAGCCGGAT |
| Sequence-based reagent | DL121_pos126_fwd | This Paper | Mutagenic PCR primer | NNSGATTACGAGCCGGATGACTG |
| Sequence-based reagent | DL121_pos127_fwd | This Paper | Mutagenic PCR primer | NNSTACGAGCCGGATGACTGG |
| Sequence-based reagent | DL121_pos128_fwd | This Paper | Mutagenic PCR primer | NNSGAGCCGGATGACTGGGAA |
| Sequence-based reagent | DL121_pos129_fwd | This Paper | Mutagenic PCR primer | NNSCCGGATGACTGGGAATCG |
| Sequence-based reagent | DL121_pos130_fwd | This Paper | Mutagenic PCR primer | NNSGATGACTGGGAATCGGTATTCAG |
| Sequence-based reagent | DL121_pos131_fwd | This Paper | Mutagenic PCR primer | NNSGACTGGGAATCGGTATTCAGC |
| Sequence-based reagent | DL121_pos132_fwd | This Paper | Mutagenic PCR primer | NNSTGGGAATCGGTATTCAGCGAATT |
| Sequence-based reagent | DL121_pos133_fwd | This Paper | Mutagenic PCR primer | NNSGAATCGGTATTCAGCGAATTCCAC |
| Sequence-based reagent | DL121_pos134_fwd | This Paper | Mutagenic PCR primer | NNSTCGGTATTCAGCGAATTCCAC |
| Sequence-based reagent | DL121_pos135_fwd | This Paper | Mutagenic PCR primer | NNSGTATTCAGCGAATTCCACGATG |
| Sequence-based reagent | DL121_pos136_fwd | This Paper | Mutagenic PCR primer | NNSTTCAGCGAATTCCACGATGC |
| Sequence-based reagent | DL121_pos137_fwd | This Paper | Mutagenic PCR primer | NNSAGCGAATTCCACGATGCTG |
| Sequence-based reagent | DL121_pos138_fwd | This Paper | Mutagenic PCR primer | NNSGAATTCCACGATGCTGATGC |
| Sequence-based reagent | DL121_pos139_fwd | This Paper | Mutagenic PCR primer | NNSTTCCACGATGCTGATGCG |
| Sequence-based reagent | DL121_pos140_fwd | This Paper | Mutagenic PCR primer | NNSCACGATGCTGATGCGCAG |
| Sequence-based reagent | DL121_pos141_fwd | This Paper | Mutagenic PCR primer | NNSGATGCTGATGCGCAGAACT |
| Sequence-based reagent | DL121_pos142_fwd | This Paper | Mutagenic PCR primer | NNSGCTGATGCGCAGAACTCTC |
| Sequence-based reagent | DL121_pos143_fwd | This Paper | Mutagenic PCR primer | NNSGATGCGCAGAACTCTCACAG |
| Sequence-based reagent | DL121_pos144_fwd | This Paper | Mutagenic PCR primer | NNSGCGCAGAACTCTCACAGC |

*Continued on next page*

*Continued*

| Reagent type (species) or resource | Designation | Source or reference | Identifiers | Additional information |
|---|---|---|---|---|
| Sequence-based reagent | DL121_pos145_fwd | This Paper | Mutagenic PCR primer | NNSCAGAACTCTCACAGCTATTGCTTTG |
| Sequence-based reagent | DL121_pos146_fwd | This Paper | Mutagenic PCR primer | NNSAACTCTCACAGCTATTGCTTTGAGATT |
| Sequence-based reagent | DL121_pos147_fwd | This Paper | Mutagenic PCR primer | NNSTCTCACAGCTATTGCTTTGAGATTCT |
| Sequence-based reagent | DL121_pos148_fwd | This Paper | Mutagenic PCR primer | NNSCACAGCTATTGCTTTGAGATTCTGG |
| Sequence-based reagent | DL121_pos149_fwd | This Paper | Mutagenic PCR primer | NNSAGCTATTGCTTTGAGATTCTGGAG |
| Sequence-based reagent | DL121_pos150_fwd | This Paper | Mutagenic PCR primer | NNSTATTGCTTTGAGATTCTGGAGCG |
| Sequence-based reagent | DL121_pos151_fwd | This Paper | Mutagenic PCR primer | NNSTGCTTTGAGATTCTGGAGCG |
| Sequence-based reagent | DL121_pos152_fwd | This Paper | Mutagenic PCR primer | NNSTTTGAGATTCTGGAGCGGC |
| Sequence-based reagent | DL121_pos153_fwd | This Paper | Mutagenic PCR primer | NNSGAGATTCTGGAGCGGCGG |
| Sequence-based reagent | DL121_pos154_fwd | This Paper | Mutagenic PCR primer | NNSATTCTGGAGCGGCGGTAA |
| Sequence-based reagent | DL121_pos155_fwd | This Paper | Mutagenic PCR primer | NNSCTGGAGCGGCGGTAACAT |
| Sequence-based reagent | DL121_pos156_fwd | This Paper | Mutagenic PCR primer | NNSGAGCGGCGGTAACATCCG |
| Sequence-based reagent | DL121_pos157_fwd | This Paper | Mutagenic PCR primer | NNSCGGCGGTAACATCCGTCG |
| Sequence-based reagent | DL121_pos158_fwd | This Paper | Mutagenic PCR primer | NNSCGGTAACATCCGTCGACAAG |
| Sequence-based reagent | DL121_pos159_fwd | This Paper | Mutagenic PCR primer | NNSTAACATCCGTCGACAAGCTTG |
| Sequence-based reagent | DL121_pos1_rev | This Paper | Mutagenic PCR primer | CGGATCCTGGCTGTGGTG |
| Sequence-based reagent | DL121_pos2_rev | This Paper | Mutagenic PCR primer | CATCGGATCCTGGCTGTG |
| Sequence-based reagent | DL121_pos3_rev | This Paper | Mutagenic PCR primer | GATCATCGGATCCTGGCTG |
| Sequence-based reagent | DL121_pos4_rev | This Paper | Mutagenic PCR primer | ACTGATCATCGGATCCTGG |
| Sequence-based reagent | DL121_pos5_rev | This Paper | Mutagenic PCR primer | CAGACTGATCATCGGATCCTG |
| Sequence-based reagent | DL121_pos6_rev | This Paper | Mutagenic PCR primer | AATCAGACTGATCATCGGATCCTG |
| Sequence-based reagent | DL121_pos7_rev | This Paper | Mutagenic PCR primer | CGCAATCAGACTGATCATCGG |
| Sequence-based reagent | DL121_pos8_rev | This Paper | Mutagenic PCR primer | CGCCGCAATCAGACTGATC |
| Sequence-based reagent | DL121_pos9_rev | This Paper | Mutagenic PCR primer | TAACGCCGCAATCAGACTGA |
| Sequence-based reagent | DL121_pos10_rev | This Paper | Mutagenic PCR primer | CGCTAACGCCGCAATCAG |
| Sequence-based reagent | DL121_pos11_rev | This Paper | Mutagenic PCR primer | TACCGCTAACGCCGCAAT |

*Continued on next page*

*Continued*

| Reagent type (species) or resource | Designation | Source or reference | Identifiers | Additional information |
|---|---|---|---|---|
| Sequence-based reagent | DL121_pos12_rev | This Paper | Mutagenic PCR primer | ATCTACCGCTAACGCCGC |
| Sequence-based reagent | DL121_pos13_rev | This Paper | Mutagenic PCR primer | GCGATCTACCGCTAACGC |
| Sequence-based reagent | DL121_pos14_rev | This Paper | Mutagenic PCR primer | AACGCGATCTACCGCTAAC |
| Sequence-based reagent | DL121_pos15_rev | This Paper | Mutagenic PCR primer | GATAACGCGATCTACCGCTAAC |
| Sequence-based reagent | DL121_pos16_rev | This Paper | Mutagenic PCR primer | GCCGATAACGCGATCTACC |
| Sequence-based reagent | DL121_pos17_rev | This Paper | Mutagenic PCR primer | CATGCCGATAACGCGATCTAC |
| Sequence-based reagent | DL121_pos18_rev | This Paper | Mutagenic PCR primer | TTCCATGCCGATAACGCG |
| Sequence-based reagent | DL121_pos19_rev | This Paper | Mutagenic PCR primer | GTTTTCCATGCCGATAACGC |
| Sequence-based reagent | DL121_pos20_rev | This Paper | Mutagenic PCR primer | GGCGTTTTCCATGCCGATAACG |
| Sequence-based reagent | DL121_pos21_rev | This Paper | Mutagenic PCR primer | CATGGCGTTTTCCATGCC |
| Sequence-based reagent | DL121_pos22_rev | This Paper | Mutagenic PCR primer | CGGCATGGCGTTTTCCAT |
| Sequence-based reagent | DL121_pos23_rev | This Paper | Mutagenic PCR primer | CCACGGCATGGCGTTTTC |
| Sequence-based reagent | DL121_pos24_rev | This Paper | Mutagenic PCR primer | GTTCCACGGCATGGCGTT |
| Sequence-based reagent | DL121_pos25_rev | This Paper | Mutagenic PCR primer | CAGGTTCCACGGCATGGC |
| Sequence-based reagent | DL121_pos26_rev | This Paper | Mutagenic PCR primer | AGGCAGGTTCCACGGCAT |
| Sequence-based reagent | DL121_pos27_rev | This Paper | Mutagenic PCR primer | GGCAGGCAGGTTCCACGG |
| Sequence-based reagent | DL121_pos28_rev | This Paper | Mutagenic PCR primer | ATCGGCAGGCAGGTTCCA |
| Sequence-based reagent | DL121_pos29_rev | This Paper | Mutagenic PCR primer | GAGATCGGCAGGCAGGTT |
| Sequence-based reagent | DL121_pos30_rev | This Paper | Mutagenic PCR primer | GGCGAGATCGGCAGGCAG |
| Sequence-based reagent | DL121_pos31_rev | This Paper | Mutagenic PCR primer | CCAGGCGAGATCGGCAGG |
| Sequence-based reagent | DL121_pos32_rev | This Paper | Mutagenic PCR primer | AAACCAGGCGAGATCGGC |
| Sequence-based reagent | DL121_pos33_rev | This Paper | Mutagenic PCR primer | TTTAAACCAGGCGAGATCGG |
| Sequence-based reagent | DL121_pos34_rev | This Paper | Mutagenic PCR primer | GCGTTTAAACCAGGCGAGAT |
| Sequence-based reagent | DL121_pos35_rev | This Paper | Mutagenic PCR primer | GTTGCGTTTAAACCAGGCGA |
| Sequence-based reagent | DL121_pos36_rev | This Paper | Mutagenic PCR primer | GGTGTTGCGTTTAAACCAGG |
| Sequence-based reagent | DL121_pos37_rev | This Paper | Mutagenic PCR primer | TAAGGTGTTGCGTTTAAACCAGG |

*Continued on next page*

*Continued*

| Reagent type (species) or resource | Designation | Source or reference | Identifiers | Additional information |
|---|---|---|---|---|
| Sequence-based reagent | DL121_pos38_rev | This Paper | Mutagenic PCR primer | ATTTAAGGTGTTGCGTTTAAACCAGG |
| Sequence-based reagent | DL121_pos39_rev | This Paper | Mutagenic PCR primer | TTTATTTAAGGTGTTGCGTTTAAACCAG |
| Sequence-based reagent | DL121_pos40_rev | This Paper | Mutagenic PCR primer | GGGTTTATTTAAGGTGTTGCGTTTAAAC |
| Sequence-based reagent | DL121_pos41_rev | This Paper | Mutagenic PCR primer | CACGGGTTTATTTAAGGTGTTGCGT |
| Sequence-based reagent | DL121_pos42_rev | This Paper | Mutagenic PCR primer | AATCACGGGTTTATTTAAGGTGTTGC |
| Sequence-based reagent | DL121_pos43_rev | This Paper | Mutagenic PCR primer | CATAATCACGGGTTTATTTAAGGTGTTG |
| Sequence-based reagent | DL121_pos44_rev | This Paper | Mutagenic PCR primer | GCCCATAATCACGGGTTTATTTAAGG |
| Sequence-based reagent | DL121_pos45_rev | This Paper | Mutagenic PCR primer | GCGGCCCATAATCACGGG |
| Sequence-based reagent | DL121_pos46_rev | This Paper | Mutagenic PCR primer | ATGGCGGCCCATAATCAC |
| Sequence-based reagent | DL121_pos47_rev | This Paper | Mutagenic PCR primer | GGTATGGCGGCCCATAATC |
| Sequence-based reagent | DL121_pos48_rev | This Paper | Mutagenic PCR primer | CCAGGTATGGCGGCCCATA |
| Sequence-based reagent | DL121_pos49_rev | This Paper | Mutagenic PCR primer | TTCCCAGGTATGGCGGCC |
| Sequence-based reagent | DL121_pos50_rev | This Paper | Mutagenic PCR primer | CGATTCCCAGGTATGGCG |
| Sequence-based reagent | DL121_pos51_rev | This Paper | Mutagenic PCR primer | GATCGATTCCCAGGTATGGCG |
| Sequence-based reagent | DL121_pos52_rev | This Paper | Mutagenic PCR primer | ACCGATCGATTCCCAGGTATG |
| Sequence-based reagent | DL121_pos53_rev | This Paper | Mutagenic PCR primer | ACGACCGATCGATTCCCA |
| Sequence-based reagent | DL121_pos54_rev | This Paper | Mutagenic PCR primer | CGGACGACCGATCGATTC |
| Sequence-based reagent | DL121_pos55_rev | This Paper | Mutagenic PCR primer | CAACGGACGACCGATCGA |
| Sequence-based reagent | DL121_pos56_rev | This Paper | Mutagenic PCR primer | TGGCAACGGACGACCGAT |
| Sequence-based reagent | DL121_pos57_rev | This Paper | Mutagenic PCR primer | TCCTGGCAACGGACGACC |
| Sequence-based reagent | DL121_pos58_rev | This Paper | Mutagenic PCR primer | GCGTCCTGGCAACGGACG |
| Sequence-based reagent | DL121_pos59_rev | This Paper | Mutagenic PCR primer | TTTGCGTCCTGGCAACGG |
| Sequence-based reagent | DL121_pos60_rev | This Paper | Mutagenic PCR primer | ATTTTTGCGTCCTGGCAAC |
| Sequence-based reagent | DL121_pos61_rev | This Paper | Mutagenic PCR primer | AATATTTTTGCGTCCTGGCAAC |
| Sequence-based reagent | DL121_pos62_rev | This Paper | Mutagenic PCR primer | GATAATATTTTTGCGTCCTGGCAAC |
| Sequence-based reagent | DL121_pos63_rev | This Paper | Mutagenic PCR primer | CAGGATAATATTTTTGCGTCCTGGC |

*Continued on next page*

*Continued*

| Reagent type (species) or resource | Designation | Source or reference | Identifiers | Additional information |
|---|---|---|---|---|
| Sequence-based reagent | DL121_pos64_rev | This Paper | Mutagenic PCR primer | GCTCAGGATAATATTTTTGCGTCCTG |
| Sequence-based reagent | DL121_pos65_rev | This Paper | Mutagenic PCR primer | TGAGCTCAGGATAATATTTTTGCGTCCT |
| Sequence-based reagent | DL121_pos66_rev | This Paper | Mutagenic PCR primer | TTGTGAGCTCAGGATAATATTTTTGCG |
| Sequence-based reagent | DL121_pos67_rev | This Paper | Mutagenic PCR primer | CGGTTGTGAGCTCAGGATAATATTTTTG |
| Sequence-based reagent | DL121_pos68_rev | This Paper | Mutagenic PCR primer | ACCCGGTTGTGAGCTCAG |
| Sequence-based reagent | DL121_pos69_rev | This Paper | Mutagenic PCR primer | CGTACCCGGTTGTGAGCT |
| Sequence-based reagent | DL121_pos70_rev | This Paper | Mutagenic PCR primer | GTCCGTACCCGGTTGTGA |
| Sequence-based reagent | DL121_pos71_rev | This Paper | Mutagenic PCR primer | ATCGTCCGTACCCGGTTG |
| Sequence-based reagent | DL121_pos72_rev | This Paper | Mutagenic PCR primer | GCGATCGTCCGTACCCGG |
| Sequence-based reagent | DL121_pos73_rev | This Paper | Mutagenic PCR primer | TACGCGATCGTCCGTACC |
| Sequence-based reagent | DL121_pos74_rev2 | This Paper | Mutagenic PCR primer | CGTTACGCGATCGTCC |
| Sequence-based reagent | DL121_pos75_rev | This Paper | Mutagenic PCR primer | CCACGTTACGCGATCGTC |
| Sequence-based reagent | DL121_pos76_rev | This Paper | Mutagenic PCR primer | CACCCACGTTACGCGATC |
| Sequence-based reagent | DL121_pos77_rev | This Paper | Mutagenic PCR primer | CTTCACCCACGTTACGCG |
| Sequence-based reagent | DL121_pos78_rev | This Paper | Mutagenic PCR primer | CGACTTCACCCACGTTACG |
| Sequence-based reagent | DL121_pos79_rev | This Paper | Mutagenic PCR primer | CACCGACTTCACCCACGT |
| Sequence-based reagent | DL121_pos80_rev | This Paper | Mutagenic PCR primer | ATCCACCGACTTCACCCA |
| Sequence-based reagent | DL121_pos81_rev | This Paper | Mutagenic PCR primer | TTCATCCACCGACTTCACC |
| Sequence-based reagent | DL121_pos82_rev | This Paper | Mutagenic PCR primer | TGCTTCATCCACCGACTTCACC |
| Sequence-based reagent | DL121_pos83_rev | This Paper | Mutagenic PCR primer | AATTGCTTCATCCACCGACTTC |
| Sequence-based reagent | DL121_pos84_rev | This Paper | Mutagenic PCR primer | CGCAATTGCTTCATCCACC |
| Sequence-based reagent | DL121_pos85_rev | This Paper | Mutagenic PCR primer | CGCCGCAATTGCTTCATC |
| Sequence-based reagent | DL121_pos86_rev | This Paper | Mutagenic PCR primer | ACACGCCGCAATTGCTTC |
| Sequence-based reagent | DL121_pos87_rev | This Paper | Mutagenic PCR primer | ACCACACGCCGCAATTGC |
| Sequence-based reagent | DL121_pos88_rev | This Paper | Mutagenic PCR primer | GTCACCACACGCCGCAAT |
| Sequence-based reagent | DL121_pos89_rev2 | This Paper | Mutagenic PCR primer | TACGTCACCACACGCC |

*Continued on next page*

*Continued*

| Reagent type (species) or resource | Designation | Source or reference | Identifiers | Additional information |
|---|---|---|---|---|
| Sequence-based reagent | DL121_pos90_rev | This Paper | Mutagenic PCR primer | TGGTACGTCACCACACGC |
| Sequence-based reagent | DL121_pos91_rev | This Paper | Mutagenic PCR primer | TTCTGGTACGTCACCACACGC |
| Sequence-based reagent | DL121_pos92_rev | This Paper | Mutagenic PCR primer | GATTTCTGGTACGTCACCACACGCC |
| Sequence-based reagent | DL121_pos93_rev | This Paper | Mutagenic PCR primer | CATGATTTCTGGTACGTCACCACACGC |
| Sequence-based reagent | DL121_pos94_rev | This Paper | Mutagenic PCR primer | CACCATGATTTCTGGTACGTCACCACA |
| Sequence-based reagent | DL121_pos95_rev | This Paper | Mutagenic PCR primer | AATCACCATGATTTCTGGTACGTCA |
| Sequence-based reagent | DL121_pos96_rev | This Paper | Mutagenic PCR primer | GCCAATCACCATGATTTCTGGTAC |
| Sequence-based reagent | DL121_pos97_rev | This Paper | Mutagenic PCR primer | GCCGCCAATCACCATGATTT |
| Sequence-based reagent | DL121_pos98_rev | This Paper | Mutagenic PCR primer | GCCGCCGCCAATCACCATG |
| Sequence-based reagent | DL121_pos99_rev | This Paper | Mutagenic PCR primer | GCGGCCGCCGCCAATCAC |
| Sequence-based reagent | DL121_pos100_rev | This Paper | Mutagenic PCR primer | AACGCGGCCGCCGCCAAT |
| Sequence-based reagent | DL121_pos101_rev | This Paper | Mutagenic PCR primer | ATAAACGCGGCCGCCGCC |
| Sequence-based reagent | DL121_pos102_rev | This Paper | Mutagenic PCR primer | TTCATAAACGCGGCCGCC |
| Sequence-based reagent | DL121_pos103_rev | This Paper | Mutagenic PCR primer | CTGTTCATAAACGCGGCC |
| Sequence-based reagent | DL121_pos104_rev | This Paper | Mutagenic PCR primer | GAACTGTTCATAAACGCGGC |
| Sequence-based reagent | DL121_pos105_rev | This Paper | Mutagenic PCR primer | CAAGAACTGTTCATAAACGCGG |
| Sequence-based reagent | DL121_pos106_rev | This Paper | Mutagenic PCR primer | TGGCAAGAACTGTTCATAAACGC |
| Sequence-based reagent | DL121_pos107_rev | This Paper | Mutagenic PCR primer | TTTTGGCAAGAACTGTTCATAAACG |
| Sequence-based reagent | DL121_pos108_rev | This Paper | Mutagenic PCR primer | CGCTTTTGGCAAGAACTGTTCATAAA |
| Sequence-based reagent | DL121_pos109_rev | This Paper | Mutagenic PCR primer | TTGCGCTTTTGGCAAGAACT |
| Sequence-based reagent | DL121_pos110_rev | This Paper | Mutagenic PCR primer | CTTTTGCGCTTTTGGCAAGAAC |
| Sequence-based reagent | DL121_pos111_rev | This Paper | Mutagenic PCR primer | AAGCTTTTGCGCTTTTGGC |
| Sequence-based reagent | DL121_pos112_rev | This Paper | Mutagenic PCR primer | ATAAAGCTTTTGCGCTTTTGGCA |
| Sequence-based reagent | DL121_pos113_rev | This Paper | Mutagenic PCR primer | CAGATAAAGCTTTTGCGCTTTTGG |
| Sequence-based reagent | DL121_pos114_rev | This Paper | Mutagenic PCR primer | CGTCAGATAAAGCTTTTGCGCTTT |
| Sequence-based reagent | DL121_pos115_rev | This Paper | Mutagenic PCR primer | ATGCGTCAGATAAAGCTTTTGCG |

*Continued on next page*

*Continued*

| Reagent type (species) or resource | Designation | Source or reference | Identifiers | Additional information |
|---|---|---|---|---|
| Sequence-based reagent | DL121_pos116_rev | This Paper | Mutagenic PCR primer | GATATGCGTCAGATAAAGCTTTTGC |
| Sequence-based reagent | DL121_pos117_rev | This Paper | Mutagenic PCR primer | GTCGATATGCGTCAGATAAAGCTTTTG |
| Sequence-based reagent | DL121_pos118_rev | This Paper | Mutagenic PCR primer | TGCGTCGATATGCGTCAGATAAA |
| Sequence-based reagent | DL121_pos119_rev | This Paper | Mutagenic PCR primer | TTCTGCGTCGATATGCGTCA |
| Sequence-based reagent | DL121_pos120_rev | This Paper | Mutagenic PCR primer | CACTTCTGCGTCGATATGCG |
| Sequence-based reagent | DL121_pos121_rev | This Paper | Mutagenic PCR primer | GTCGATGTTCTCGGCGGT |
| Sequence-based reagent | DL121_pos122_rev | This Paper | Mutagenic PCR primer | GCCGTCGATGTTCTCGGC |
| Sequence-based reagent | DL121_pos123_rev | This Paper | Mutagenic PCR primer | GTCGCCGTCGATGTTCTC |
| Sequence-based reagent | DL121_pos124_rev | This Paper | Mutagenic PCR primer | GGTGTCGCCGTCGATGTT |
| Sequence-based reagent | DL121_pos125_rev | This Paper | Mutagenic PCR primer | ATGGGTGTCGCCGTCGAT |
| Sequence-based reagent | DL121_pos126_rev | This Paper | Mutagenic PCR primer | GAAATGGGTGTCGCCGTC |
| Sequence-based reagent | DL121_pos127_rev | This Paper | Mutagenic PCR primer | CGGGAAATGGGTGTCGCC |
| Sequence-based reagent | DL121_pos128_rev | This Paper | Mutagenic PCR primer | ATCCGGGAAATGGGTGTC |
| Sequence-based reagent | DL121_pos129_rev | This Paper | Mutagenic PCR primer | GTAATCCGGGAAATGGGTGTC |
| Sequence-based reagent | DL121_pos130_rev | This Paper | Mutagenic PCR primer | CTCGTAATCCGGGAAATGGG |
| Sequence-based reagent | DL121_pos131_rev | This Paper | Mutagenic PCR primer | CGGCTCGTAATCCGGGAA |
| Sequence-based reagent | DL121_pos132_rev | This Paper | Mutagenic PCR primer | ATCCGGCTCGTAATCCGG |
| Sequence-based reagent | DL121_pos133_rev | This Paper | Mutagenic PCR primer | GTCATCCGGCTCGTAATCC |
| Sequence-based reagent | DL121_pos134_rev | This Paper | Mutagenic PCR primer | CCAGTCATCCGGCTCGTA |
| Sequence-based reagent | DL121_pos135_rev | This Paper | Mutagenic PCR primer | TTCCCAGTCATCCGGCTC |
| Sequence-based reagent | DL121_pos136_rev | This Paper | Mutagenic PCR primer | CGATTCCCAGTCATCCGG |
| Sequence-based reagent | DL121_pos137_rev | This Paper | Mutagenic PCR primer | TACCGATTCCCAGTCATCCG |
| Sequence-based reagent | DL121_pos138_rev | This Paper | Mutagenic PCR primer | GAATACCGATTCCCAGTCATCC |
| Sequence-based reagent | DL121_pos139_rev | This Paper | Mutagenic PCR primer | GCTGAATACCGATTCCCAGTC |
| Sequence-based reagent | DL121_pos140_rev | This Paper | Mutagenic PCR primer | TTCGCTGAATACCGATTCCCA |
| Sequence-based reagent | DL121_pos141_rev | This Paper | Mutagenic PCR primer | GAATTCGCTGAATACCGATTCCC |

*Continued on next page*

*Continued*

| Reagent type (species) or resource | Designation | Source or reference | Identifiers | Additional information |
|---|---|---|---|---|
| Sequence-based reagent | DL121_pos142_rev | This Paper | Mutagenic PCR primer | GTGGAATTCGCTGAATACCGATTC |
| Sequence-based reagent | DL121_pos143_rev | This Paper | Mutagenic PCR primer | ATCGTGGAATTCGCTGAATACC |
| Sequence-based reagent | DL121_pos144_rev | This Paper | Mutagenic PCR primer | AGCATCGTGGAATTCGCTG |
| Sequence-based reagent | DL121_pos145_rev | This Paper | Mutagenic PCR primer | ATCAGCATCGTGGAATTCGC |
| Sequence-based reagent | DL121_pos146_rev | This Paper | Mutagenic PCR primer | CGCATCAGCATCGTGGAATT |
| Sequence-based reagent | DL121_pos147_rev | This Paper | Mutagenic PCR primer | CTGCGCATCAGCATCGTG |
| Sequence-based reagent | DL121_pos148_rev | This Paper | Mutagenic PCR primer | GTTCTGCGCATCAGCATC |
| Sequence-based reagent | DL121_pos149_rev | This Paper | Mutagenic PCR primer | AGAGTTCTGCGCATCAGC |
| Sequence-based reagent | DL121_pos150_rev | This Paper | Mutagenic PCR primer | GTGAGAGTTCTGCGCATCAG |
| Sequence-based reagent | DL121_pos151_rev | This Paper | Mutagenic PCR primer | GCTGTGAGAGTTCTGCGC |
| Sequence-based reagent | DL121_pos152_rev | This Paper | Mutagenic PCR primer | ATAGCTGTGAGAGTTCTGCG |
| Sequence-based reagent | DL121_pos153_rev | This Paper | Mutagenic PCR primer | GCAATAGCTGTGAGAGTTCTGC |
| Sequence-based reagent | DL121_pos154_rev | This Paper | Mutagenic PCR primer | AAAGCAATAGCTGTGAGAGTTCTG |
| Sequence-based reagent | DL121_pos155_rev | This Paper | Mutagenic PCR primer | CTCAAAGCAATAGCTGTGAGAGTTC |
| Sequence-based reagent | DL121_pos156_rev | This Paper | Mutagenic PCR primer | AATCTCAAAGCAATAGCTGTGAGAGTT |
| Sequence-based reagent | DL121_pos157_rev | This Paper | Mutagenic PCR primer | CAGAATCTCAAAGCAATAGCTGTGAG |
| Sequence-based reagent | DL121_pos158_rev | This Paper | Mutagenic PCR primer | CTCCAGAATCTCAAAGCAATAGCTG |
| Sequence-based reagent | DL121_pos159_rev | This Paper | Mutagenic PCR primer | CCGCTCCAGAATCTCAAAGC |
| Sequence-based reagent | DL121_E154R_F | This Paper | Mutagenic PCR primer | ctctcacagctattgctttaggattctggagcggcggtaa |
| Sequence-based reagent | DL121_E154R_R | This Paper | Mutagenic PCR primer | ttaccgccgctccagaatcctaaagcaatagctgtgagag |
| Sequence-based reagent | DL121_D122W_F | This Paper | Mutagenic PCR primer | gtaatccgggaaatgggtccagccgtcgatgttctcggc |
| Sequence-based reagent | DL121_D122W_R | This Paper | Mutagenic PCR primer | gccgagaacatcgacggctggacccatttcccggattac |
| Sequence-based reagent | DL121_D127W_F | This Paper | Mutagenic PCR primer | cagtcatccggctcgtaccacgggaaatgggtgtcgc |
| Sequence-based reagent | DL121_D127W_R | This Paper | Mutagenic PCR primer | gcgacacccatttcccgtggtacgagccggatgactg |
| Sequence-based reagent | DL121_M16A_F | This Paper | Mutagenic PCR primer | cggcatggcgttttccgcgccgataacgcgatct |
| Sequence-based reagent | DL121_M16A_R | This Paper | Mutagenic PCR primer | agatcgcgttatcggcgcggaaaacgccatgccg |

*Continued on next page*

*Continued*

| Reagent type (species) or resource | Designation | Source or reference | Identifiers | Additional information |
|---|---|---|---|---|
| Sequence-based reagent | DL121_A9N_F | This Paper | Mutagenic PCR primer | catgccgataacgcgatctacatttaacgccgcaatcagactgatc |
| Sequence-based reagent | DL121_A9N_R | This Paper | Mutagenic PCR primer | gatcagtctgattgcggcgttaaatgtagatcgcgttatcggcatg |
| Sequence-based reagent | DL121_R52K_F | This Paper | Mutagenic PCR primer | tcctggcaacggcttaccgatcgattcccaggtatggc |
| Sequence-based reagent | DL121_R52K_R | This Paper | Mutagenic PCR primer | gccatacctgggaatcgatcggtaagccgttgccagga |
| Sequence-based reagent | DL121_E120P_F | This Paper | Mutagenic PCR primer | ctagagtggtggccagtggcacttctgcgtcgatat |
| Sequence-based reagent | DL121_E120P_R | This Paper | Mutagenic PCR primer | atatcgacgcagaagtgccactggccaccactctag |
| Sequence-based reagent | DL121_S148C_F | This Paper | Mutagenic PCR primer | aagcaatagctgtgacagttctgcgcatcagcatc |
| Sequence-based reagent | DL121_S148C_R | This Paper | Mutagenic PCR primer | gatgctgatgcgcagaactgtcacagctattgctt |
| Sequence-based reagent | DL121_H124Q_F | This Paper | Mutagenic PCR primer | tcgtaatccgggaactgggtgtcgccgtc |
| Sequence-based reagent | DL121_H12RQ_R | This Paper | Mutagenic PCR primer | gacggcgacacccagttcccggattacga |
| Sequence-based reagent | DL121_D27N_F | This Paper | Mutagenic PCR primer | aaaccaggcgagattggcaggcaggttcc |
| Sequence-based reagent | DL121_D27N_R | This Paper | Mutagenic PCR primer | ggaacctgcctgccaatctcgcctggttt |
| Sequence-based reagent | DL121_D87A_F | This Paper | Mutagenic PCR primer | catgatttctggtacggcaccacacgccgcaat |
| Sequence-based reagent | DL121_D87A_R | This Paper | Mutagenic PCR primer | attgcggcgtgtggtgccgtaccagaaatcatg |
| Sequence-based reagent | Thrombin_to_TEV_F | This Paper | Mutagenic PCR primer | cttccagggtcatgggatgatgatcagtctgattgc |
| Sequence-based reagent | Thrombin_to_TEV_R | This Paper | Mutagenic PCR primer | tacaggttctcaccaccgtggtggtggtg |
| Sequence-based reagent | DL121_SL1V2_F | This Paper | Round one Amplicon PCR primer | cactctttccctacacgacgctcttccga tctnnnnatcaccatcatcaccacagc |
| Sequence-based reagent | DL121_SL1V2_R | This Paper | Round one Amplicon PCR primer | tgactggagttcagacgtgtgctcttcc gatctnnnnaccgatcgattcccaggta |
| Sequence-based reagent | DL121_SL2V2_F | This Paper | Round one Amplicon PCR primer | cactctttccctacacgacgctcttccga tctnnnngcaacaccttaaataaacccg |
| Sequence-based reagent | DL121_SL2V2_R | This Paper | Round one Amplicon PCR primer | tgactggagttcagacgtgtgctcttccga tctnnnngatttctggtacgtcaccaca |
| Sequence-based reagent | DL121_SL3V2_F | This Paper | Round one Amplicon PCR primer | cactctttccctacacgacgctcttccga tctnnnngtaacgtgggtgaagtcg |
| Sequence-based reagent | DL121_SL3V2_R | This Paper | Round one Amplicon PCR primer | tgactggagttcagacgtgtgctcttccga tctnnnnctcgatgcgctctagagtg |
| Sequence-based reagent | DL121_SL4V2_F | This Paper | Round one Amplicon PCR primer | cactctttccctacacgacgctcttccga tctnnnnaagaagaccgccgagaacat |
| Sequence-based reagent | DL121_SL4V2_R | This Paper | Round one Amplicon PCR primer | tgactggagttcagacgtgtgctcttcc gatctnnnncttaagcattatgcggccg |
| Sequence-based reagent | DL121_CLV3_F | This Paper | Round one Amplicon PCR primer | cactctttccctacacgacgctcttccga tctnnnngacacccatttcccggattacgagc |
| Sequence-based reagent | DL_WTTS_R3 | This Paper | Round one Amplicon PCR primer | tgactggagttcagacgtgtgctcttccga tctnnnngccgtgtacaatacgattactttctg |

*Continued on next page*

*Continued*

| Reagent type (species) or resource | Designation | Source or reference | Identifiers | Additional information |
|---|---|---|---|---|
| Sequence-based reagent | D501 | Illumina/Reynolds et al. Cell 2011 [20] | Round two Amplicon PCR primer | aatgatacggcgaccaccgagatctacac tatagcctacactctttccctacacgac |
| Sequence-based reagent | D502 | Illumina/Reynolds et al. Cell 2011 [20] | Round two Amplicon PCR primer | aatgatacggcgaccaccgagatctac acatagaggcacactctttccctacacgac |
| Sequence-based reagent | D503 | Illumina/Reynolds et al. Cell 2011 [20] | Round two Amplicon PCR primer | aatgatacggcgaccaccgagatcta caccctatcctacactctttccctacacgac |
| Sequence-based reagent | D504 | Illumina/Reynolds et al. Cell 2011 [20] | Round two Amplicon PCR primer | aatgatacggcgaccaccgagatctaca cggctctgaacactctttccctacacgac |
| Sequence-based reagent | D505 | Illumina/Reynolds et al. Cell 2011 [20] | Round two Amplicon PCR primer | aatgatacggcgaccaccgagatctacaca ggcgaagacactctttccctacacgac |
| Sequence-based reagent | D506 | Illumina/Reynolds et al. Cell 2011 [20] | Round two Amplicon PCR primer | aatgatacggcgaccaccgagatctacac taatcttaacactctttccctacacgac |
| Sequence-based reagent | D507 | Illumina/Reynolds et al. Cell 2011 [20] | Round two Amplicon PCR primer | aatgatacggcgaccaccgagatctaca ccaggacgtacactctttccctacacgac |
| Sequence-based reagent | D508 | Illumina/Reynolds et al. Cell 2011 [20] | Round two Amplicon PCR primer | aatgatacggcgaccaccgagatctaca cgtactgacacactctttccctacacgac |
| Sequence-based reagent | D701 | Illumina/Reynolds et al. Cell 2011 [20] | Round two Amplicon PCR primer | caagcagaagacggcatacgagatc gagtaatgtgactggagttcagacgtg |
| Sequence-based reagent | D702 | Illumina/Reynolds et al. Cell 2011 [20] | Round two Amplicon PCR primer | caagcagaagacggcatacgagattct ccggagtgactggagttcagacgtg |
| Sequence-based reagent | D703 | Illumina/Reynolds et al. Cell 2011 [20] | Round two Amplicon PCR primer | caagcagaagacggcatacgagataa tgagcggtgactggagttcagacgtg |
| Sequence-based reagent | D704 | Illumina/Reynolds et al. Cell 2011 [20] | Round two Amplicon PCR primer | caagcagaagacggcatacgagatggaa tctcgtgactggagttcagacgtg |
| Sequence-based reagent | D705 | Illumina/Reynolds et al. Cell 2011 [20] | Round two Amplicon PCR primer | caagcagaagacggcatacgagatttct gaatgtgactggagttcagacgtg |
| Sequence-based reagent | D706 | Illumina/Reynolds et al. Cell 2011 [20] | Round two Amplicon PCR primer | caagcagaagacggcatacgagatac gaattcgtgactggagttcagacgtg |
| Sequence-based reagent | D707 | Illumina/Reynolds et al. Cell 2011 [20] | Round two Amplicon PCR primer | caagcagaagacggcatacgagatagctt caggtgactggagttcagacgtg |
| Sequence-based reagent | D708 | Illumina/Reynolds et al. Cell 2011 [20] | Round two Amplicon PCR primer | caagcagaagacggcatacgagatgc gcattagtgactggagttcagacgtg |
| Sequence-based reagent | D709 | Illumina/Reynolds et al. Cell 2011 [20] | Round two Amplicon PCR primer | caagcagaagacggcatacgagatca tagccggtgactggagttcagacgtg |
| Sequence-based reagent | D710 | Illumina/Reynolds et al. Cell 2011 [20] | Round two Amplicon PCR primer | caagcagaagacggcatacgagatttc gcggagtgactggagttcagacgtg |
| Sequence-based reagent | D711 | Illumina/Reynolds et al. Cell 2011 [20] | Round two Amplicon PCR primer | caagcagaagacggcatacgagatgcgc gagagtgactggagttcagacgtg |
| Sequence-based reagent | D712 | Illumina/Reynolds et al. Cell 2011 [20] | Round two Amplicon PCR primer | caagcagaagacggcatacgagatctatc gctgtgactggagttcagacgtg |
| Commercial assay or kit | QuikChange II site-directed mutagenesis kit | Agilent | Cat. #: 200523 | |
| Software, algorithm | usearch v11.0.667 | Edgar Bioinformatics 2010 (PMID:20709691) | Merge read pairs | https://www.drive5.com/usearch/ |

## Experimental model and subject details

### *Escherichia coli* expression and selection strains

ER2566 Δ*folA* Δ*thyA E. coli* were used for all growth in vivo growth rate measurements; this strain was a kind gift from Dr. Steven Benkovic and is the same used in *Reynolds et al., 2011* and *Thompson et al., 2020* (*Reynolds et al., 2011*; *Thompson et al., 2020*). XL1-Blue *E. coli* (genotype: *recA1 endA1 gyrA96 thi-1 hsdR17 supE44 relA1 lac* [F′ *proAB lacI*q*ZΔM15* Tn*10*(Tet$^r$)]) from Agilent

Technologies were used for cloning, mutagenesis, and plasmid propagation. BL21(DE3) *E. coli* (genotype: *fhuA2 [lon] ompT gal (λ DE3) [dcm] ΔhsdS. λ DE3 = λ sBamHIo ΔEcoRI-B int::(lacI::Pla-cUV5::T7 gene1) i21 Δnin5*) from New England Biolabs were used for protein expression.

## Method details

### DHFR saturation mutagenesis library construction

The construction of the DHFR-LOV2 saturation mutagenesis library was done as described in *Thompson et al., 2020* (*Thompson et al., 2020*). Four sublibraries were generated to cover the entire mutational space of *E. coli* DHFR: positions 1–40 (sublibrary1, SL1), positions 41–80 (sublibrary2, SL2), positions 81–120 (sublibrary3, SL3), and positions 121–159 (sublibrary4, SL4) Inverse PCR with NNS mutagenic primers (N = A/T/G/C, S = G/C) was done at every position in DHFR to produce all amino acid substitution. The vector with DHFR-LOV2 121 and TYMS in a pACYC-Duet vector was described in *Reynolds et al., 2011* (*Reynolds et al., 2011*).

The NNS primers were phosphorylated with T4 polynucleotide kinase (NEB, cat#M0201S). 20 µL phosphorylations was prepared according to the following recipe: 16.5 µL sterile water, 2 µL T4 ligase buffer, 0.5 µL T4 PNK enzyme, and 1 µL 100 µM NNS primers. The reactions were then heated at 37°C for 1 hr and 65°C for 20 min.

PCR reactions were set up using 2x Q5 mastermix (NEB, cat#M0492), 10 ng of plasmid template, and 500 nM forward and reverse primers. PCR was performed in the following steps: (1) 98°C for 30 s, (2) 98°C for 10 s, (3) 55°C for 30 s, (4) 72°C for 2 min, (5) return to step 2 for 22 cycles, (6) 72°C for 5 min. 25 µL of PCR reaction was mixed with 1 µL of DpnI (NEB, cat#R0176) at 37°C for 4 hr. The samples were then purified by gel extraction and a DNA Clean and Concentrator −5 kit (Zymo Research, cat#D4014). PCR product solution were then phosphorylated with a second round of T4 PNK: 100 µL of gel-extracted PCR product, 12 µL of 10x T4 ligase buffer, 5 µL of T4 PNK, 5 µL of sterile water and were incubated at 37°C for 1 hr with 90°C for 30 s. The reactions were ligated with 100 µL PNK phosphorylated PCR product, 15 µL T4 ligase (NEB, cat#M0202S), 30 µL T4 ligase buffer and, 155 µL sterile water. The reaction was incubated at room temperature for 24 hr.

The concentration of each reaction was quantified by gel densitometry (ImageJ) and combined in equimolar ratios to form sublibraries. The library was divided up into four sublibraries with sublibrary 1 covering positions 1–40, sublibrary 2 covering positions 41–80, sublibrary 3 covering positions 81–120, and sublibrary 4 covering positions 121–150. Sublibraries were transformed into electrocompetent XL1-Blue *E. coli* using a MicroPulser Electroporator (Bio Rad) and gene pulser cuvettes (Bio Rad, cat#165–2089). Cultures were miniprepped using a GeneJET plasmid miniprep kit (Thermo Scientific, cat#K05053). Library completeness was verified by deep sequencing on a MiSeq (Illumina).

### Growth rate measurements in the turbidostat for DHFR DL121 mutant library

DHFR DL121 sublibraries were transformed into ER2566 ΔfolA ΔthyA *E. coli* by electroporation using a MicroPulser Electroporator (Bio Rad) and gene pulser cuvettes (Bio Rad, cat#165–2089). Cultures were grown overnight at 37°C in GM9 minimal media (93.0 mM Sodium (Na$^+$), 22.1 mM Potassium (K$^+$), 18.7 mM Ammonium (NH$_4$), 1.0 mM Calcium (Ca$^{2+}$), 0.1 mM Magnesium (Mg$^{2+}$), 29.2 mM Chloride (Cl$^-$), 0.1 mM Sulfate (SO$_4^{2-}$), and 42.2 mM Phosphate (PO$_4^{3-}$), 0.4% glucose) pH 6.50, containing 50 µg/mL thymidine and 30 µg/mL chloramphenicol (Sigma, cat#C0378-5G) as well as folA mix which contains 38 µg/mL glycine (Sigma, cat#50046), 75.5 µg/mL L-methionine (Sigma, cat#M9625) 1 µg/mL calcium pantothenate (Sigma, cat#C8731), and 20 µg/mL adenosine (Sigma, cat#A9251). Four hours before the start of the experiment, the overnight culture was diluted to an optical density of 0.1 at 600 nm in GM9 minimal media containing 50 µg/mL thymidine and 30 µg/mL chloramphenicol and incubated for four hours at 30°C. The cultures were centrifuged at 2000 RCF for 10 min and resuspended in the experimental conditions of GM9 minimal media containing 1 µg/mL thymidine and 30 µg/mL chloramphenicol. This was repeated two more times. The cultures were then back-diluted to an OD600 of 0.1 in 16 mL/vial of media. The turbidostat described in *Toprak et al., 2013* was used in continuous culture (turbidostat) mode with a clamp OD600 of 0.15 and a temperature of 30°C. Each vial had a stir bar. Vials designated as 'lit' had one 5V blue LED active. The optical density was continuously monitored throughout the experiment. 1 mL samples were taken at the beginning of selection (0 hr) and at 4, 8, 12, 16, 20, and 24 hr into

selection and were centrifuged at 21,130 RCF for 5 min at room temperature with the pellet being stored at −20°C for sequencing sample preparation.

## Growth rate measurements in the turbidostat for DHFR control library

Wild-type DHFR, 12 DHFR point mutants (D27N, F31V, F31Y, F31Y-L54I, G121V, G121V-F31Y, G121V-M42F, L54I, L54I-G121V, M42F, and W22H), and three chimeric DHFR-LOV2 fusion constructs (DL116, DL121, and DL121-C450S) each in a pACYC-Duet vector with TYMS as described in *Reynolds et al., 2011* were transformed into ER2566 Δ*folA* Δ*thyA E. coli* by electroporation using a MicroPulser Electroporator (Bio Rad) and gene pulser cuvettes (Bio Rad, cat#165–2089) (*Reynolds et al., 2011*). Cultures were grown overnight at 37°C in GM9 minimal media (93.0 mM Sodium ($Na^+$), 22.1 mM Potassium ($K^+$), 18.7 mM Ammonium ($NH_4$), 1.0 mM Calcium ($Ca^{2+}$), 0.1 mM Magnesium ($Mg^{2+}$), 29.2 mM Chloride ($Cl^-$), 0.1 mM Sulfate ($SO_4^{2-}$), and 42.2 mM Phosphate ($PO_4^{3-}$), 0.4% glucose) pH 6.50, containing 50 μg/mL thymidine and 30 μg/mL chloramphenicol (Sigma, cat#C0378-5G) as well as folA mix which contains 38 μg/mL glycine (Sigma, cat#50046), 75.5 μg/mL L-methionine (Sigma, cat#M9625) 1 μg/mL calcium pantothenate (Sigma, cat#C8731), and 20 μg/mL adenosine (Sigma, cat#A9251). Four hours before the start of the experiment the overnight culture was diluted to an optical density of 0.1 at 600 nm in GM9 minimal media containing 50 μg/mL thymidine and 30 μg/mL chloramphenicol and incubated for four hours at 30°C. The cultures were centrifuged at 2000 RCF for 10 min and resuspended in the experimental conditions of GM9 minimal media containing 1 μg/mL thymidine and 30 μg/mL chloramphenicol. This was repeated two more times. The cultures were then back-diluted to an OD600 of 0.1 and pooled at equal (1/16th) ratios and aliquoted into four 'dark' and four 'lit' vials with 16 ml culture. The turbidostat described in *Toprak et al., 2013* was used in continuous culture (turbidostat) mode with a clamp OD600 of 0.15 and a temperature of 30°C. Each vial had a stir bar. Vials designated as 'lit' had one 5V blue LED active. The optical density was continuously monitored throughout the experiment. One mL samples were taken at the beginning of selection (0 hr) and at 4, 8, 12, 16, 20, and 24 hr into selection and were centrifuged at 21,130 RCF for 5 min at room temperature with the pellet being stored at −20°C for sequencing sample preparation.

## Plate reader assay for *E. coli* growth

Single point mutant DHFR-D27N, DL121 chimeric protein, and DL121 with a point mutant D27N each in a pACYC-Duet vector with TYMS as described in *Reynolds et al., 2011* were transformed into ER2566 Δ*folA* Δ*thyA E. coli* by electroporation using a MicroPulser Electroporator (Bio Rad) and gene pulser cuvettes (Bio Rad, cat#165–2089) (*Reynolds et al., 2011*). Cultures were grown overnight at 37°C in GM9 minimal media (93.0 mM Sodium ($Na^+$), 22.1 mM Potassium ($K^+$), 18.7 mM Ammonium ($NH_4$), 1.0 mM Calcium ($Ca^{2+}$), 0.1 mM Magnesium ($Mg^{2+}$), 29.2 mM Chloride ($Cl^-$), 0.1 mM Sulfate ($SO_4^{2-}$), and 42.2 mM Phosphate ($PO_4^{3-}$), 0.4% glucose) pH 6.50, containing 50 μg/mL thymidine and 30 μg/mL chloramphenicol (Sigma, cat#C0378-5G) as well as folA mix which contains 38 μg/mL glycine (Sigma, cat#50046), 75.5 μg/mL L-methionine (Sigma, cat#M9625) 1 μg/mL calcium pantothenate (Sigma, cat#C8731), and 20 μg/mL adenosine (Sigma, cat#A9251). Four hours before the start of the experiment, the overnight culture was diluted to an optical density of 0.1 at 600 nm in GM9 minimal media containing 50 μg/mL thymidine and 30 μg/mL chloramphenicol and incubated for four hours at 30°C. The cultures were centrifuged at 2000 RCF for 10 min and resuspended in the experimental conditions of GM9 minimal media containing either 0, 1, or 50 μg/mL thymidine and 30 μg/mL chloramphenicol. The cells were centrifuged and resuspended two more times. The cultures were then back-diluted to an OD600 of 0.005 into 96-well plates with six replicates each.

## Next-generation sequencing Amplicon sample preparation

Cell pellets were lysed by the addition of 10 μL sterile water, mixed by pipetting, and incubated at 98°C for 5 min. One μL of this was then combined with 5 μL Q5 buffer (NEB, cat#M0491S), 0.5 μL 10 mM DNTP (Thermo Scientific, cat#R0192), 2.5 μL of 10 mM forward and reverse primers specific to the sublibrary and containing the TruSeq adapter sequence (Appendix 1: SL1V2, SL2V2, SL3V2, SL4V2, DL121CLV3F, and DL_WTTS_R3), 0.25 μL of Q5 enzyme (NEB, cat#M0491S) and 13.25 μL of sterile water. These samples were then heated at 98°C for 90 s and then cycled through 98°C for 10 s 63–65°C (sublibrary 1: 66°C, sublibrary 2: 63°C, sublibrary 3: 64°C, and sublibrary 4: 65°C) for 15 s

and then 72℃ for 15 s, repeating 20 times with a final 72℃ heating for 120 s in a Veriti 96-well ther-mocycler (Applied Biosystems). These samples were then amplified using TruSeq PCR reactions with a unique combination of i5/i7 indexing primers for each timepoint. 1 µL of this PCR reaction was then combined with 5 µL Q5 buffer (NEB, cat#M0491S), 0.5 µL 10 mM DNTP (Thermo Scientific, cat#R0192), 2.5 µL of 10 mM forward and reverse primers, 0.25 µL of Q5 enzyme (NEB, cat#M0491S) and 13.25 µL of sterile water. These samples were then heated at 98℃ for 30 s and then cycled through 98℃ for 10 s 55℃ for 10 s and then 72℃ for 15 s, repeating 20 times with a final 72℃ heat-ing for 60 s in a Veriti 96 well thermocycler (Applied Biosystems). Amplified DNA from i5/i7 PCR reaction was quantified using the picogreen assay (Thermo Scientific, cat#P7589) on a Victor X3 mul-timode plate reader (Perkin Elmer) and the samples were mixed in an equimolar ratio. The DNA was then purified by gel extraction and a DNA Clean and Concentrator −5 kit (Zymo Research, cat#D4014). DNA quality was determined by 260 nm/230 nm and 260 nm/280 nm ratios on a DS-11 +spectrophotometer (DeNovix) and concentration was determined using the Qubit 3 (Thermo Scientific). Pooled samples were sent to GeneWiz where they were analyzed by TapeStation (Agilent Technologies) and sequenced on a HiSeq 4000 sequencer (Illumina) with 2 × 150 bp dual index run with 30% PhiX spike-in yielding 1.13 billion reads. The control library was sequenced in-house using a MiSeq sequencer (Illumina) with 2 × 150 bp dual index 300 cycle MiSeq Nano Kit V2 (Illumina cat#15036522) with 20% PhiX (Illumina cat#FC-110–3001) spike-in yielding 903,488 reads.

## DHFR chimeric expression constructs

The *E. coli* DHFR LOV2 fusion was cloned as an NcoI/XhoI fragment into the expression vector pHIS8-3 (*Lee et al., 2008*; *Reynolds et al., 2011*). Point mutants were engineered into the DHFR gene using QuikChange II site-directed mutagenesis kits (Agilent cat#200523) using primers speci-fied in Appendix 1. All DHFR/LOV2 fusions for purification were expressed under control of a T7 promoter, with an N-terminal 8X His-tag for nickel affinity purification. The existing thrombin cleav-age site (LVPRGS) following the His-tag in pHIS8-3 was changed to a TEV cleavage site using restric-tion-free PCR to improve the specificity of tag removal (*Bond and Naus, 2012*). All constructs were verified by Sanger DNA sequencing.

## Protein expression and purification

DHFR-LOV2 chimeric proteins were expressed in BL21(DE3) *E. coli* grown at 30℃ in Terrific Broth (12 g/L Tryptone, 24 g/L yeast extract, 4 mL/L glycerol, 17 mM $KH_2PO_4$, and 72 mM $K_2HPO_4$). Pro-tein expression was induced when the cells reached an absorbance at 600 nm of 0.7 with 0.25 mM IPTG, and cells were grown at 18℃ overnight. Cell pellets were lysed by sonication in binding buffer (500 mM NaCl, 10 mM imidazole, 50 mM Tris-HCL, pH 8.0) added at a volume of 5 ml/g cell pellet. Next the lysate was clarified by centrifugation and the soluble fraction was incubated with equili-brated Ni-NTA resin (Qiagen cat#4561) for 1 hr at 4℃. After washing with one column volume of wash buffer (300 mM NaCl, 20 mM imidazole, 50 mM Tris-HCL, pH 8.0) the DHFR-LOV2 protein was eluted with elution buffer (1M NaCl, 250 mM imidazole, 50 mM Tris-HCL, pH 8.0) at 4℃. Eluted pro-tein was dialyzed into dialysis buffer (300 mM NaCl, 1% glycerol, 50 mM Tris-HCl, pH 8.0) at 4℃ overnight in 10,000 MWCO Thermo protein Slide A Lyzer (Fisher Scientific cat#PI87730). Following dialysis, the protein was then purified by size exclusion chromatography (HiLoad 16/600 Superdex 75 pg column, GE Life Sciences cat#28989333). Purified protein was concentrated using Amicon Ulta 10 k M.W. cutoff concentrator (Sigma cat#UFC801024) and flash frozen using liquid $N_2$ prior to enzymatic assays.

## Steady state Michaelis Menten measurements

The protein was spun down at 21,130 RCF at 4℃ for 10 min and the supernatant was moved to a new tube with any pellet being discarded. The concentration of the protein was quantitated by A280 using a DS-11 +spectrophotometer (DeNovix) with an extinction coefficient of 44920 mM$^{-1}$ cm$^{-1}$. The parameters $k_{cat}$ and $K_m$ under Michaelis-Menten conditions were determined by measur-ing the initial velocity for the depletion of NADPH as measured in absorbance at 340 nm, with an extinction coefficient of 13.2 mM$^{-1}$ cm$^{-1}$. This is done in a range of substrate concentrations with a minimum of 8 data points around 4 $K_m$, 2 $K_m$, 1.5 $K_m$, $K_m$, 0.8 $K_m$, 0.5 $K_m$, 0.25 $K_m$ and 0. The initial velocities (slope of the first 15 s) were plotted vs. the concentration of Dihydrofolate and fit to a

Michaelis Menten model using non-linear regression in GraphPad Prism 7. The reactions are run in MTEN buffer (50 mM 2-(N-morpholino)ethanesulfonic acid, 25 mM tris base, 25 mM ethanolamine, 100 mM NaCl) pH 7.00, 5 mM Dithiothreitol, 90 µM NADPH (Sigma-Aldrich cat#N7505) quantitated by A340. Dihydrofolate (Sigma-Aldrich cat#D7006) is suspended in MTEN buffer pH 7.00 with 0.35% β-mercaptoethanol and quantitated by A282 with an extinction coefficient of 28 mM$^{-1}$ cm$^{-1}$. Depletion of NADPH is observed in 1 mL cuvettes with a path length of 1 cm in a Lambda 650 UV/VIS spectrometer (Perkin Elmer) with attached water Peltier system set to 17°C. Lit samples are illuminated for at least 2 min by full spectrum 125 watt 6400K compact fluorescent bulb (Hydrofarm Inc cat#FLC125D). Dark samples were also exposed to the light in the same way as the lit samples but were in opaque tubs. Velocity, $V = k_{cat}[P]\frac{[S]}{K_M+[S]}$, was calculated using the concentration of DHF found in wild-type *E. coli* (~25 µM **Kwon et al., 2008**).

## Spectrophotometry of the LOV2 chromophore

The spectra of the LOV2 chromophore is determined with a Lambda 650 UV/VIS spectrometer (Perkin Elmer) at 350–550 nm using paired 100 µL Hellma ultra micro cuvettes (Sigma cat#Z600350-1EA) with a path length of 1 cm. Purified protein in was diluted (when possible) to 20 µM in MTEN buffer pH 7.00 with 0.35% β-mercaptoethanol The lit samples are illuminated for at least 2 min by full spectrum 125 watt 6400K compact fluorescent bulb (hydrofarm Inc). Relaxation of the lit state chromophore is observed in the Lambda 650 UV/VIS spectrometer (Perkin Elmer) at 447 nm (dark peak) using paired 100 µL Hellma ultra micro cuvettes (Sigma cat#Z600350-1EA) with a path length of 1 cm.

## Quantification and statistical analysis

### Next-generation sequencing

The sequencing data analysis can be divided into two portions: (1) Read Joining, Filtering and Counting, followed by (2) Calculating Relative Fitness and Final Filtering. We describe each step below; all code was implemented in Bash shell scripting or Python 3.6.4. All analysis codes have been made available as a series of python 3 Jupyter Notebooks on github (https://github.com/reynoldsk/allostery-in-dhfr; **McCormick et al., 2021**; copy archived at swh:1:rev:dd8ee13f775f8b08548d64868f15e46583cbf543).

### Read joining, filtering, and counting

The data analysis began with unjoined illumina fastq.gz files separated by index (generated by Gene-Wiz). The forward and reverse reads were combined using usearch v11.0.667 using the i86linux32 package. The commands given to usearch are contained in the script UCOMBINER.bsh.

Reads of each paired fastq file are identified and quality checked using the script DL121_fastq_analysis.py. Mutant nucleotide counts and number of wild-type reads are stored in a dictionary where the read count is separated by file name (vial and timepoint eg: T2V3) and sublibrary. If any nucleotide in the coding region is below a qscore cutoff of 30, that read is discarded. Counts of every nucleotide are saved in a text file by timepoint and vial.

Converting nucleotide variation to amino acid count as well as probabilistic sequencer error correction is done by the Hamming_analysis.ipynb script. Given the probabilistic nature of base calling on the Illumina platform, one can expect a number of reads that were errantly called. For each codon, the expected number of reads due to sequencing noise was calculated with the formula:

$$\mathbf{NErrant_t^{Mut}} = \mathbf{N_t^{WT}}\left(10^{\left(\frac{\mu \mathbf{Q}}{-10}\right)}\right)^{\mathbf{HD}}$$

The number of errant mutants ($\mathbf{NErrant_t^{Mut}}$) can be calculated from the number of observed wild type ($\mathbf{N_t^{WT}}$), the average Q score of the sequencing run $\mu\mathbf{Q}$, and the hamming distance (**HD**) or number of mutations away from. The number of errant mutants then subtracted from the actual mutant count. In addition to the number of observed wild type, this is calculated for every possible mutation observed, up to the 31 other nucleotide codons, (NNK codons are discarded due to the nature of library construction). Once the total number of errant reads are calculated and subtracted from the mutant and wild-type counts, they are then converted into the amino acid sequence and are saved

into text files. These files are then used to load information for calculation of growth rate and allostery.

## Calculating relative fitness and final filtering

Growth_Rate_and_Allostery.ipynb
was the python script used for this analysis. Relative frequency was calculated as follows:

$$f(t) = \ln\left(\frac{N_t^{Mut}/N_t^{Wt}}{N_{t=0}^{Mut}/N_{t=0}^{Wt}}\right)$$

Variant frequencies ($N_t^{Mut}$) were determined relative to WT ($N_t^{Wt}$) and normalized to the initial frequency distribution at t=0. The relative growth rate then calculated by linear regression of these normalized frequencies. Light dependence was calculated as the difference between lit vs. dark growth rates. Variant frequency was only calculated if there were more than 50 mutant reads at time zero. Definitions for sector identity, conservation values, and surface identity used in SectorSurfaceDefinitions.ipynb are the same as those from *Reynolds et al., 2011*. Accessible surface area was calculated using MSMS, using a probe size of 1.4Å and excluding water as well as heteroatoms (*Sanner et al., 1996*). Values for total surface areas were taken from *Chothia, 1976*. Together these were used to calculate relative solvent accessible surface area, and 25% was used as a cutoff for 'surface'. A surface site is considered to contact the sector if the atoms comprising the peptide bond contact *any* sector atoms. Contact is defined as the sum of the atom's Pauling radii + 20%.

To determine significant allosteric mutations, a p-value for each mutation was computed by unequal variance t-test under the null hypothesis that the lit and dark replicate measurements have equal means. Two cutoffs were used, a standard cutoff of p=0.05, and a more stringent cutoff that is adjusted to consider multiple hypothesis testing. A multiple-hypothesis testing adjusted p-value of p=0.016 was determined by Sequential Goodness of Fit (*Carvajal-Rodriguez and de Uña-Alvarez, 2011*). General analysis and figures made from this data are performed in allostery_analysis.ipynb.

## Acknowledgements

The authors are grateful to Dr. Tanja Kortemme for facilitating our collaboration with Samuel Thompson. We also acknowledge Dr. Elliott Ross and Dr. Rama Ranganathan for thoughtful discussion and feedback. We thank Christine Ingle for her assistance with DHFR purification and kinetics protocols, and other members of the Reynolds lab for comments on the manuscript and discussions throughout the development of this work. FUNDING This work was supported by NSF Grant # 1942354 to KAR, and in part by the Gordon and Betty Moore Foundation's Data Driven Discovery Initiative through grant GBMF4557 to KAR.

## Additional information

### Funding

| Funder | Grant reference number | Author |
|---|---|---|
| National Science Foundation | CAREER Award 1942354 | Kimberly A Reynolds |
| Gordon and Betty Moore Foundation | Data Driven Discovery Initiative GBMF4557 | Kimberly A Reynolds |

The funders had no role in study design, data collection and interpretation, or the decision to submit the work for publication.

### Author contributions

James W McCormick, Conceptualization, Formal analysis, Investigation, Methodology, Writing - original draft, Writing - review and editing; Marielle AX Russo, Investigation, Methodology, Writing - review and editing; Samuel Thompson, Resources, Investigation, Methodology, Writing - review and editing; Aubrie Blevins, Investigation, Writing - review and editing; Kimberly A Reynolds,

Conceptualization, Formal analysis, Supervision, Funding acquisition, Methodology, Writing - original draft, Writing - review and editing

## Author ORCIDs
James W McCormick [ORCID] https://orcid.org/0000-0001-7573-2300
Kimberly A Reynolds [ORCID] https://orcid.org/0000-0003-4805-0317

## Decision letter and Author response
Decision letter https://doi.org/10.7554/eLife.68346.sa1
Author response https://doi.org/10.7554/eLife.68346.sa2

## Additional files

### Supplementary files
• Supplementary file 1. Supplementary tables. (**a**) Steady state kinetic parameters for select point mutants of the DL121 fusion. The parameter $k_{cat}$ is reported in units of $s^{-1}$, $K_m$ is in units of μM. Error is calculated as standard error of the mean over three replicates. Related to *Figure 4* of the main text. (**b**) Fisher Exact Test p-values for the null hypothesis that the sector and inactivating mutants are independent properties. Inactivating mutations are defined as those that yield relative growth rates at or below the growth rate for DL121-D27N. Over a range of sector definitions, the null hypothesis is rejected at a confidence level of 0.05 or better, shown in red. Sector definitions were taken from *Reynolds et al., 2011* (*Rivoire et al., 2016*). (**c**) Fisher Exact Test p-values for the null hypothesis that conserved positions and inactivating mutants are independent. Calculations were made over a range of conservation definitions chosen to result in an equal number positions as the sector positions in Supplementary file 1b 23, 36, 40, and 49 positions respectively. In all cases, the null hypothesis is rejected at a confidence level of 0.05 or better (red), and inactivating mutations are enriched at conserved positions beyond expectation due to random chance. Conservation values are calculated as in *Reynolds et al., 2011* (*Rivoire et al., 2016*), and reflect the Kullback-Leibler relative entropy of amino acid frequencies at each DHFR position. (**d**) Fisher Exact Test p-values for the null hypothesis that the sector and allosteric mutations are independent. We compared over four sector cutoffs (as defined in *Rivoire et al., 2016*) and at two cutoffs for allostery significance (a standard p-value of 0.05, and an adjusted p-value of 0.016). The multiple hypothesis testing adjusted p-value was obtained by Sequential Goodness of Fit (SGoF, *Carvajal-Rodriguez and de Uña-Alvarez, 2011*). The top table shows the association between sector positions and allostery enhancing mutations; the bottom table computes the associate between sector positions and allostery disrupting mutations. In nearly all cases, the null hypothesis is rejected at a confidence level of 0.05 or better, shown in red. (**e**) Fisher Exact Test p-values for the null hypothesis that the solvent accessible DHFR surface and allosteric mutations are independent. At two cutoffs for allostery (a standard p-value of 0.05, and an adjusted p-value of 0.016), the null hypothesis is rejected at a confidence level of 0.05 or better, shown in red. (**f**) Statistical association of allosteric mutations and surface positions that are either within or contacting the sector. Contacting was defined as two atoms within the sum of their Pauling radii plus 20%. A surface site contacts the sector if the peptide bond atoms of the surface site contact any atoms in the sector position. P-values were computed by Fisher exact test with the null hypothesis that the sector and allosteric mutations are independent. Cutoffs for sector definition as defined in *Rivoire et al., 2016* are shown as well as mutants determined to effect allostery either at a 95% confidence interval (p<0.05) or at the multiple hypothesis testing adjusted p-value (p<0.016). The null hypothesis that there is no relationship between allosteric mutations and sector or sector contacting positions on the surface of DHFR of the DL121 chimera is rejected at a confidence level of 0.05 or better over a range of cutoffs, shown in red. Allostery enhancing mutations are depleted from sector connected surface sites, while allostery disrupting mutations are enriched (in comparison with random expectation). (**g**) Statistical association of allosteric mutations and surface positions that are contacting the sector. In contrast to Supplementary file 1f, surface positions within the sector are excluded. Fisher Exact Test p-values were calculated for the null hypothesis that the sector and allosteric mutations are independent. Cutoffs for sector definition as defined in *Rivoire et al., 2016* are shown as well as mutants determined to effect allostery either at

a 95% confidence interval (p<0.05) or at the multiple hypothesis testing adjusted p-value (p<0.016). At most cutoff combinations, there is not a statistically significant association between sector connected surface sites and either mutations that enhance (top panel) or disrupt allostery (bottom panel).

- Transparent reporting form

### Data availability

Sequencing data (resulting from amplicon sequencing) have been deposited in the NCBI SRA under BioProject: PRJNA706683. All analysis codes have been made available as a series of python 3 Jupyter Notebooks on github: https://github.com/reynoldsk/allostery-in-dhfr (copy archived at https://archive.softwareheritage.org/swh:1:rev:dd8ee13f775f8b08548d64868f15e46583cbf543).

The following dataset was generated:

| Author(s) | Year | Dataset title | Dataset URL | Database and Identifier |
|---|---|---|---|---|
| McCormick JW, Russo MAX, Thompson S, Blevins A, Reynolds KA | 2021 | Effect of saturation mutagenesis to novel allosteric system on allosteric effect | https://www.ncbi.nlm.nih.gov/bioproject/PRJNA706683 | NCBI BioProject, PRJNA706683 |

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
