## [Decision Letter]

**Acceptance summary:**

The authors use an innovative experimental system to study how mutations can influence allosteric protein regulation. They fuse a light sensitive protein to an enzyme and examine how light can become a regulator of the enzyme. They show that allostery-improving mutations can occur on the surface of the enzyme. These experiments illustrate how allosteric regulation can evolve and be improved by mutations following protein domain fusion. This work is of broad interest for scientists interested in the structural basis of allostery, how it can be engineered into proteins and how it evolves.

**Decision letter after peer review:**

Thank you for submitting your article "Structurally distributed surface sites tune allosteric regulation" for consideration by *eLife*. Your article has been reviewed by 3 peer reviewers, and the evaluation has been overseen by a Reviewing Editor and Michael Marletta as the Senior Editor. The following individuals involved in review of your submission have agreed to reveal their identity: Georg Hochberg (Reviewer #1); Adrian Serohijos (Reviewer #2).

The reviewers have discussed their reviews with one another, and the Reviewing Editor has drafted this to help you prepare a revised submission. You will see below changes that are recommended by the reviewers and that I would like to request. Some are comments regarding the text, missing information in the figures or supplementary figures that would better fit in the main text. Another one regards the claim about the additive effects of the mutations. The reviewers felt that the experiments or measurements were not designed to test this so it may be better to only discuss this point rather than be affirmative about the lack of non-additivity. Finally, there are few comments on results interpretation and comparison with other, already existing experiments that would help improve the manuscript if dealt with.

*Reviewer #1:*

McCormick et al. present a very interesting study of how mutations can affect allosteric regulation of proteins. This phenomenon is of interest to biochemists, synthetic biologists, and evolutionary biochemists. Their approach uses a synthetic fusion between an enzyme and a light regulated domain that makes the enzyme weakly regulated by light. James W. McCormick et al. perform a deep mutational single site scan to test whether single mutations in the enzyme domain can enhance or disrupt allostery. They find a small but appreciable fraction that can enhance allostery and that these mutations can in some cases be additive. These findings suggest how allosteric regulation might evolve after domain fusion: by a sequence of small effect, potentially additive substitutions. There is currently very little data about how exactly allostery evolves, so this is a valuable study and of interest to a wide audience.

There are three main conclusions that I judge to be of wide interest.

1) That allostery improving sites fall outside of so-called 'sectors'. These interconnected sets of sites, usually derived from statistical coupling analysis, have been proposed to represent allosteric networks. This works is an interesting challenge to that idea. By a well know practitioner of SCA, no less.

2) That allostery improving mutations are enriched on surface sites and have comparatively small effect sizes.

3) That at least some allostery improving mutations can act additively to produce large dynamic allosteric ranges when introduced together.

Together these data suggest that allostery can be tuned through a gradual evolutionary trajectory, in which many individually small effect steps are taken to make a protein allosteric. This view would be consistent with how early proponents of the modern synthesis would have thought about the emergence of complex traits in general.

The study is in general carefully carried out and the interpretations appropriate and interesting as far as I can tell. I find the system used to study this question quite innovative. Though I will admit that I am no expert in analyzing deep mutational scanning data. I therefore cannot comment with authority on the appropriateness of the analyses that generate fitness values from sequencing data..But based on what I can reasonably judge the major conclusions are supported.

As caveats I can only note that of course this system is somewhat artificial, but the authors are aware of this. They correctly note that it simulates what might happen directly after a domain fusion event that introduces a base level of allostery. Another criticism may be that near-additivity is only shown for a group of three mutations, so the generality of this observation is somewhat unclear. But studies on the evolvability of allostery are still so rare that this is nonetheless a valuable contribution.

I am a bit confused about exactly what correlates with growth rates. The authors initially show that k_cat_/K_m_ correlates very well with relative growth rate. When they authors then do experiments on the library in the dark and lit state, instead only the k_cat_ value seems to matter. I was confused about this difference and didn't really understand their explanation of it. I'm pretty sure this is just a writing issue and would encourage the authors just to re-read the relevant sections with fresh eyes.

I was also unable to locate the E145R and S148C mutants in Figure 4C. They are indicated as having been characterized in vitro, is there a reason there were left out of the correlation?

*Reviewer #2:*

Allosteric regulation enables the activity of one site on a protein to modulate function at another spatially distinct site. This work by McCormick et al. studied the spectrum of mutational effects on allosteric regulation by deep mutational scanning of a well-established system of DHFR-LOV2 fused synthetic construct (Dihydrofolate reductase/Light-oxygen-voltage-sensing domain). Major strengths of this work are clear statement of objectives (which mutations improve, decrease or tune an allosteric system and by how much) and well-designed approaches. As in any deep mutational scan study, the crux is in the validity of the selection system. The DHFR-LOV2 construct was previously developed by the last author (Reynolds, Cell 2011) to demonstrate the existence of "protein sectors" or networks of physically contiguous and coevolving amino acids that relay allosteric regulation signals. In addition, the group recently published a plasmid-based based deep mutational scan of DHFR using a folA and thyA knock-out *E. coli* (Thompson *eLife* 2020). In essence, by combining the approaches of these two earlier papers, they evaluated the effects of ~1540 single mutations on allostery. These high-throughput measurements were validated and found to be in excellent agreement with 11 DHFR point mutations whose allosteric effects were measured in isolation. Using robust statistical cut-offs, they found that only ~4.5% (69) of mutations significantly influence allostery. Mutations improving allostery are distributed and enriched on the protein surface, while mutations disrupting allostery are enriched in the LOV2 insertion site. The latter is perhaps not surprising, since LOV2 insertion at this site could have perturbed residue contacts in DHFR crucial for allosteric regulation. This also provides a valuable lesson that synthetic-coupled domains can be furthered engineered at the fusion site to improve regulatory control.

It remains to be seen whether the findings from DHFR-LOV2 construct is generalizable to other proteins or to protein complexes that evolved naturally. But overall, this is a significant work towards our mechanistic understanding not only on the structural basis of protein allostery, but also of its evolution and potential optimization and design.

1) The authors claim that allosteric effects of mutations are additive (non-epistatic), which is quite profound and unintuitive. However, in there data, there is a difference between measured and expectations from log-additivity of the double (M16A,H124Q) and triple (M16A,D87A,H124Q) mutants (Figure 6B). This difference suggest epitasis for allostery. The authors may want to clarify this apparent discrepancy.

2) An interesting result is the stronger effect of allostery on k_cat_ but not on K_m_. The authors surmise this from the in vivo concentration of DHF in *E. coli*, which is ~25 μm and well above the measured K_m_. In the selection system, the proteins are expressed on plasmids, presumably at concentrations higher than endogenous in vivo concentrations of DHFR. Might the protein abundance due to expression on plasmid affect this particular result?

3) Since there is a prior DHFR deep mutational scan (Thompson *eLife* 2020), which estimated the effects of mutations in vivo activity. I believe it would have been enlightening to correlate and compare those results with the mutational effects measured for allostery. This could shed light on the pleiotropic effects of mutations on protein properties (e.g., activity vs. allostery).

4) Overall, the manuscript is written logically and very clearly. However, the authors should consider promoting Supplementary Figure S4 as a panel in Figure 3 or Figure 4. The supplementary figure provides a compelling picture of the "allosteric" landscape of DHFR.

5) Reference 19 is not shown correctly.

*Reviewer #3:*

The authors report four key findings: 1) A highly quantitative high-throughput assay for this enzyme's function; 2) Relatively few mutants affected allostery, and those that did had a small effect; 3) Allostery-disrupting mutations tended to be near the active site; 4) Allostery-enhancing mutations tended to be on the surface.

On the whole, their conclusions were amply justified. The collected data were extraordinarily clean, and the analysis refreshingly transparent. The authors do an excellent job supporting each of their claims with evidence. The manuscript is well-written and the figures are clear. The authors do a good job in their supplement of showing the material, controls, and methods necessary to reproduce their work.

The work is well-done, but not flashy. The effect-sizes were small and the structural patterns of the mutational effects relatively weak. The sheer quality of the work, however, makes it an important and interesting addition to the literature. Further, the structural pattern of allostery-enhancement on the surface and not in "sectors" detected by co-evolution was intriguing and somewhat unexpected.

1. I think the manuscript could benefit from a discussion of context. The number of allostery-altering mutations and effect-sizes are small. Is this unexpected? The authors note that their results could come from the fact that they are in an artificial, unoptimized system. But what sorts of numbers/magnitudes would they expect in a naturally evolved or optimized protein?

2. I also felt there was a relative paucity of structural rationale for their findings. Do the authors have any hypotheses for why surface positions would be uniquely good at enhancing allostery in this context? They note several other studies that found unconserved sites were a bit better at tuning allostery than conserved. Does their new work provide any insights (or hints of insights) as to why this would hold?

---

## [Author Response]

*Reviewer #1:*

[…]

I am a bit confused about exactly what correlates with growth rates. The authors initially show that k_cat_/K_m_ correlates very well with relative growth rate. When the authors then do experiments on the library in the dark and lit state, instead only the k_cat_ value seems to matter. I was confused about this difference and didn't really understand their explanation of it. I'm pretty sure this is just a writing issue and would encourage the authors just to re-read the relevant sections with fresh eyes.

We would like to thank the reviewer for raising this important point. Our control library consists of well characterized DHFR point mutants chosen to span a range of values for both *k_cat_* (0.05-79.2 s^-1^) and K_m_ (330-1.1 µM). Somewhat by serendipity, the assumption that all of these enzyme mutations operate in a first order kinetics regime where *k_cat_* and K_m_ contribute equally to growth rate (in the plot of growth rate vs catalytic power) yielded a high correlation.

However, the DL121 fusion construct and all characterized mutations have a K_m_ less than 2, far below the reported in vivo DHF concentration of 25µM in *E. coli* (Kwon et al., 2008, Nat Chem Biol v4:602-608). If these values are accurate in vivo, we would expect the DHFR domain of our chimera to operate under a pseudo-zero-order regime where perturbations to *k_cat_* are disproportionately or entirely responsible for alterations in growth rate. Indeed, the correlation plots in Figure 4—figure supplement 6 lend evidence to this explanation. As reviewer 1 has rightly identified, this has resulted in confusion and serves to add a layer of difficulty for the reader.

To rectify this we have simplified the main text to use only velocity (V = V_max_[S]/(K_m_ + [S])) at 25µM DHF to compare with growth rate, and we now use the ratio in lit/dark velocity to quantify the allosteric effect. We have altered figures 1D, 2C, 4D, 6B, as well as the text to reflect this change. The original version of Figure 4D has been moved to the supplement (Figure 4—figure supplement 6). This change allows us to relate growth rate to a consistent biochemical measure in all figures.

In using velocity to describe our data, we have incorporated two assumptions: 1) we presume minimal variation in protein abundance between mutants (enzyme concentration is equal to one) and 2) we fix the substrate concentration at 25 µM. These assumptions are described in the main text, on page 7, lines 140-146. The strong correlation between velocity and our experimentally measured growth rates (Figure 2C) indicates these assumptions are reasonable. Additionally, the observed correlation between the allosteric effect as measured in vitro and in vivo (Figure 4D) is robust to variation in the concentration of DHF above ~5 µM. Thus, the precise choice of substrate concentration is not critical to our claims (see Author response image 1). Given that presenting our results in terms of velocity allows for a more consistent presentation of the data, we have chosen to go forward with these assumptions.

**Author response image 1. sa1fig1:** 

I was also unable to locate the E145R and S148C mutants in Figure 4C. They are indicated as having been characterized in vitro, is there a reason there were left out of the correlation?

While E154R and S148C were selected for in vitro characterization, we were unable to purify active protein at quantities needed for kinetics. We found that E154R and S148C fell out of solution under several different purification strategies. Given that E154R is a high confidence allostery-enhancing mutation in our in vivo experiments, we opted to report our failed attempt at purification as the observed instability is suggestive of potential allosteric mechanism. In response to the concern raised by reviewer 1, we have expanded our discussion of these mutants on page 14, line 264 as well as mentioning them in the figure legend of 4B.

Page 14 – lines 267-271:

Modified from: “We expressed and purified the selected DL121 mutants to near homogeneity; S148C and E154R did not yield sufficient quantities of active protein for in vitro studies.”

To: “We expressed and purified the selected DL121 mutants to near homogeneity; S148C and E154R did not yield sufficient quantities of active protein for in vitro studies. We find it noteworthy that E154R – one of the strongest allostery-enhancing mutations in vivo – was unstable in multiple purification strategies.”

Updated figure legend 4B to include:

“S148C and E154R did not yield sufficient quantities of active protein for further in vitro characterization.”

*Reviewer #2:*

[…]

1) The authors claim that allosteric effects of mutations are additive (non-epistatic), which is quite profound and unintuitive. However, in there data, there is a difference between measured and expectations from log-additivity of the double (M16A,H124Q) and triple (M16A,D87A,H124Q) mutants (Figure 6B). This difference suggest epitasis for allostery. The authors may want to clarify this apparent discrepancy.

We agree that additivity in the allosteric effects of single mutations is unintuitive. However, when considering error in the measurements, the difference between the log-additive expectation (given single mutation data) and the experimentally observed allosteric effect for the mutant combinations is statistically insignificant. This was true whether we considered the ratio of lit and dark *k_cat_* values (as in the original Figure 6B), or the ratio of lit and dark velocities (as in the updated Figure 6B). For clarity we have added the phrase: “There is not a statistically significant difference between the expected and observed allosteric effects (p = 0.07 for M16A,H124Q, p=0.48 for M16A,D87A,H124Q; as computed by unpaired t-test).” To the figure legend of Figure 6B.

2) An interesting result is the stronger effect of allostery on k_cat_ but not on K_m_. The authors surmise this from the in vivo concentration of DHF in E. coli, which is ~25 μm and well above the measured K_m_. In the selection system, the proteins are expressed on plasmids, presumably at concentrations higher than endogenous in vivo concentrations of DHFR. Might the protein abundance due to expression on plasmid affect this particular result?

Here, we understand the reviewer is asking about potential explanations for the fact that all of the biochemically characterized mutations modulated allostery through *k_cat_* rather than K_m_. In the manuscript, we suggest this could be a consequence of in vivo experimental conditions that reflect pseudo-zero-order kinetics ([DHF] >> K_m_). In this scenario, K_m_ has little impact on enzyme velocity (and thus growth rate), and we would expect that our assay would predominantly detect mutations effecting *k_cat_*. This idea is loosely supported by the fact that the in vivo concentration of DHF in wildtype *E. coli* is 25 µM, an order of magnitude above the measured K_m_ for the characterized DL121 mutations (~1 µM).

However, as the reviewer correctly points out, our system differs from wild type *E. coli* in several important ways: 1) the DL121 chimera has much-reduced activity relative to native DHFR, 2) it is expressed on a low-copy plasmid using a non-native promoter, and 3) the plasmid also contains the gene encoding thymidylate synthase, a folate metabolic enzyme which produces DHF (*thyA* is deleted from the chromosome in our selection strain). It is difficult to intuit what the combined effect of these changes on DHF concentration might be, and quite possible that the in vivo concentration of DHF varies from the 25 µM measurement made for wild type *E. coli*… potentially even in a mutant-specific fashion. Nevertheless, the strong correlation between DHFR velocity (calculated at 25 µM DHF) and relative growth rate suggests that 25 µM is a reasonable choice (new Figure 2C). Importantly, we observe that the correlation between allosteric effect (as measured in vivo) and the ratio of lit:dark enzyme velocities (as determined in vitro) is stable across a broad range of intracellular DHF concentrations (>5 µM, see also the response to reviewer 1), so our results do not strongly depend on a specific choice of DHF concentration. Thus, we feel our original explanation is plausible, though the present data do not conclusively demonstrate it.

Alternatively, it could be that the allosteric mutations predominantly impact *k_cat_* for biophysical reasons. Perhaps the placement of the LOV2 domain makes it more energetically feasible for light to modulate *k_cat_* than K_m_. Our result that distributed surface mutations modulate allosteric regulation requires additional mechanistic explanation, and this is something we hope to address in future work. We now include a more developed and nuanced explanation of these ideas on page 14, line 285 – page 15, line 336:

“So why might the characterized allosteric mutations predominantly effect k_cat_? One plausible explanation is that the conditions of our in vivo experiments fall within a pseudo-zero-order kinetics regime ([DHF] >> K_m_). […] In any case, the 1.3 to 2 fold changes in k_cat_ translate to similar fold changes in enzyme velocity”.

3) Since there is a prior DHFR deep mutational scan (Thompson eLife 2020), which estimated the effects of mutations in vivo activity. I believe it would have been enlightening to correlate and compare those results with the mutational effects measured for allostery. This could shed light on the pleiotropic effects of mutations on protein properties (e.g., activity vs. allostery).

In Thompson et al., the authors (which partly overlap with those of this manuscript) investigated how the presence of Lon protease reshapes the single mutation fitness landscape of DHFR. The selection conditions for that experiment were chosen to resolve changes in catalytic activity near WT DHFR, while the conditions of the present work (McCormick et al) were chosen to resolve changes in activity near the less-active DL121 chimera. Accordingly, these experiments differ in the stringency of selection in two ways: (1) McCormick et al. included 1 µg/mL thymidine supplementation in the media, while Thompson et al. did not (2) In the McCormick et al. experiments, DHFR is translated from a Shine-Delgarno consensus ribosome binding site (RBS) with high translation initiation rate, while the construct in Thompson et al. makes use of an RBS that is predicted to have 0.05x translation rate (by the RBS calculator, Salis et al., 2009, Nat Biotech v27:946). Beyond these differences in experimental setup, the LOV2 insertion in DL121 renders the DHFR domain less active, and potentially less stable. For these reasons, a strong relationship between the two datasets is not expected.

Nonetheless, we very much agree with the reviewer that it is important to have a look at the two datasets together and see what can be learned. We scaled each deep mutational scanning dataset by the appropriate reference WT growth rate (unmutated DL121 for McCormick et al., WT DHFR for Thompson et al) to facilitate comparison. In the following plot, zero represents non-growing mutations, while one represents WT-like growth. DL121 mutations classified as slow-growing are not shown (growth rate < DL121 D27N).

The data reveal that DL121 was far less robust to mutation than native DHFR. Native DHFR exhibited a peak of relative growth rates near one, with a tail of deleterious mutations. In contrast, the distribution of fitness effects for DL121 was strongly skewed towards deleterious mutations with no observable peak near one. Indeed, many of the mutations that were deleterious to DL121 activity displayed growth rates near WT in the context of native DHFR (lower right quadrant of Author response image 2).

**Author response image 2. sa1fig2:** 

Importantly, the set of null mutations for each experiment (those mutations which were too deleterious to fit a reliable growth rate) show statistically significant overlap (p-value = 0.0002), indicating that the mutations yielding gross disruptions to activity are largely the same. There was no relationship between DHFR mutations that significantly altered Lon protease sensitivity and DL121 mutations that significantly altered allostery (p-value = 0.826). Due to the significant differences in the model system, experimental design, and limited comparability between the two sets of experiments, we have refrained from including a comparison and a discussion of Thompson et al. in the main text.

4) Overall, the manuscript is written logically and very clearly. However, the authors should consider promoting Supplementary Figure S4 as a panel in Figure 3 or Figure 4. The supplementary figure provides a compelling picture of the "allosteric" landscape of DHFR.

In reflection, we agree and have implemented this suggestion. The heatmap of allosteric effects was moved from (former) Supplemental Figure 4C to Figure 4A and the text has been accordingly updated.

5) Reference 19 is not shown correctly.

This has been fixed. Thank you for bringing this to our attention.

*Reviewer #3:*

[…] 1. I think the manuscript could benefit from a discussion of context. The number of allostery-altering mutations and effect-sizes are small. Is this unexpected? The authors note that their results could come from the fact that they are in an artificial, unoptimized system. But what sorts of numbers/magnitudes would they expect in a naturally evolved or optimized protein?

The number and effect size of allosteric mutations in naturally evolved systems still remains relatively uncharacterized. Quantifying the *frequency* of allostery-altering mutations requires a comprehensive single mutation study. We are aware of only two near-comprehensive mutagenesis experiments that have directly measured the allosteric effects of mutations in native proteins. The first is an alanine scan of human liver pyruvate kinase wherein allosteric effect was characterized in vitro (Tang and Fenton, 2017, Hum Mutat. 38:1132). The second is the pioneering work of Jeffrey Miller on lac repressor (Kleina et al., 1990, JMB 212:295, Markiewicz et al., 1994, JMB 240:421, Suckow et al., 1996, JMB 261:509). In the lac repressor studies, mutations that impact allostery are categorized with those that fail to bind inducer, so it is difficult to distill out the separate roles of mutations. For pyruvate kinase, only ~5% of mutations effected allosteric inhibition by alanine, while nearly ~30% impacted allosteric activation by fructose-1,6-bisphosphate. Given this, it remains difficult to define a general expectation for *how many* mutations tune allostery in an evolved system. However, the results on pyruvate kinase suggest it might be higher than what we observed for DL121.

These same studies also provide some insight into mutational effect sizes, and suggest that effect sizes in natural proteins can be larger than what we observed here. Tang et al. write that “most mutations that increased allosteric inhibition by alanine only did so marginally”, though they did identify a single mutation that increased alanine inhibition by six-fold. They also reported a single mutation of pyruvate kinase that increased allosteric activation by 22-fold. With further study – including deep mutational scans of natural allosteric systems – we should gain a better appreciation of how our results (in a synthetic system) compare to more optimized or evolved systems.

To provide this additional context, we have added a reference to Tang and Fenton, on page 22 – lines 447-450:

“While we observed that the number of allosteric mutations is few and the effect sizes are generally small, a previous study of allostery tuning mutations in pyruvate kinase indicated that up to 30% of mutations can tune allostery, with the maximum effect size approaching 22-fold [54].”

2. I also felt there was a relative paucity of structural rationale for their findings. Do the authors have any hypotheses for why surface positions would be uniquely good at enhancing allostery in this context? They note several other studies that found unconserved sites were a bit better at tuning allostery than conserved. Does their new work provide any insights (or hints of insights) as to why this would hold?

Yes, we agree with the reviewer – we intentionally provide only limited structural rationale for our findings. While the experiments here map the extent, distribution, and strength of allosteric mutations on an unoptimized allosteric system, they do not provide information on the mechanism by which this allosteric signal is modulated. Beyond briefly mentioning a possible role for solvent and the protein hydration layer (p.24, lines 485-487), we have refrained from commenting about the structural mechanism of allostery from a concern of unsupported claims and misrepresenting our findings. While a mechanistic analysis of these mutations is out of the scope of this work, we strongly agree our findings now demand biophysical explanation. This is an active area of interest and the subject of ongoing research in our laboratory.